

# Large ensemble assessment of the Arctic stratospheric polar vortex

Ales Kuchar[1,*], Maurice Öhlert[1], Roland Eichinger[2,3], and Christoph Jacobi[1]

[1]Leipzig Institute for Meteorology, Faculty of Physics and Earth Sciences, Leipzig University, Germany
[2]Deutsches Zentrum für Luft- und Raumfahrt (DLR), Institut für Physik der Atmosphäre, Oberpfaffenhofen, Germany
[3]Charles University Prague, Faculty of Mathematics and Physics, Department of Atmospheric Physics, Prague, Czech
Republic
[*]now at: University of Natural Resources and Life Sciences (BOKU), Institute of Meteorology and Climatology, Vienna,
Austria

**Correspondence:** A. Kuchar (ales.kuchar@boku.ac.at)

**Abstract.** The stratospheric polar vortex (SPV) is a phenomenon comprising strong westerly winds during winter in both hemispheres. Especially in the Northern Hemisphere (NH) the SPV is highly variable and is frequently disrupted by sudden stratospheric warmings (SSWs). SPV dynamics are relevant because of both ozone chemistry and its impact on tropospheric dynamics. In this study, we evaluate the capability of climate models to simulate the NH SPV by comparing large ensembles
of historical simulations to the ERA5 reanalysis data. For this, we analyze geometric-based diagnostics at 3 pressure levels that describe SPV morphology. Moreover, we assess the ability of the models to simulate SSWs subdivided into SPV split and displacement events. A rank histogram analysis reveals that no model exactly reproduces ERA5 in all diagnostics at all levels. Concerning SPV aspect ratio and centroid latitude, most models are biased to some extent, but the strongest deviations can be found for the kurtosis. Some models underestimate the variability of the SPV area. Assessing the reliability of the ensembles
in distinguishing SPV displacement and split events, we find large differences between the model ensembles. In general, SPV displacements are represented better than splits in the simulation ensembles, and high-top models and models with finer vertical resolution perform better. A good performance in representing the geometric-based diagnostics in rank histograms is found to be not necessarily connected to a good performance in simulating displacements and splits. Understanding the biases and improving the representation of SPV dynamics in climate model simulations can help to improve credibility of climate
projections, in particular with focus on polar stratospheric dynamics and ozone.

## 1 Introduction

In winter the dynamics of the mid-latitude and polar stratosphere are dominated by the stratospheric polar vortex (SPV). The SPV is a circumpolar band of usually strong westerly winds. It forms in autumn due to the cooling of the polar strato-sphere. When the stratosphere warms again in spring, the temperature gradient reverses and easterly winds prevail during
summer (Holton, 1980). The SPV affects the concentration of ozone over the poles: strong winds are accompanied by lower than average temperatures, that allow the formation of polar stratospheric clouds, where ozone depleting substances are acti-vated (Langematz et al., 2014; Lawrence et al., 2020). Via stratosphere-troposphere coupling (Baldwin and Dunkerton, 2001), the SPV can influence tropospheric circulation patterns, temperatures, and precipitation (Thompson et al., 2002; Butler et al.,



2017; King et al., 2019). Due to the impact of the stratospheric circulation on the surface weather, uncertainties associated with

25 SPV relates to tropospheric uncertainties in climate projections, in particular the windiness over Europe, winter precipitation in the Mediterranean region (Scaife et al., 2012; Zappa and Shepherd, 2017) and sea level pressure over the Arctic (Simpson et al., 2018). Especially in the Northern Hemisphere (NH), the SPV is highly variable (Baldwin et al., 2021), the strongest changes happen during so-called sudden stratospheric warmings (SSWs). SSWs are abrupt warmings of the stratosphere, connected with a zonal wind reduction or even reversal during a so-called major SSW. In the NH, SSWs occur on average about 6 times

30 per decade (Charlton and Polvani, 2007). They are much less frequent in the Southern Hemisphere, with only one recorded major SSW in 2002 since the beginning of the satellite era (Jucker et al., 2021). To distinguish a SSW from the final warming when the wind direction changes to East for the entire summer, an SSW warming must be followed by at least 10 consecutive days of westerly winds (Butler et al., 2015).

SSWs can be categorized into two different kinds. Either the SPV is split into two separate SPVs or it is displaced to lower

latitudes (Charlton and Polvani, 2007). It is still a matter of current research whether the pressure patterns before the event, and especially the surface pressure response after the event are different depending on the type of SSW (Mitchell et al., 2013; Seviour et al., 2013; Maycock and Hitchcock, 2015). Particularly for exceptionally strong SPV conditions, dynamical downward coupling from the stratosphere to the troposphere can be observed even influencing surface weather and climate patterns (Black, 2002; Scaife et al., 2005; Kidston et al., 2015; Baldwin and Dunkerton, 2001). The skill of weather forecasts in the

extratropical troposphere is enhanced following an SSW (Sigmond et al., 2013; Tripathi et al., 2015).

Multiple studies have focused on analyzing whether climate change alters stratospheric dynamics (Manzini et al., 2014; Ayarzagüena et al., 2020; Rao and Garfinkel, 2021). In the model simulations of the Climate Model Intercomparison Project 5 (CMIP5), the largest uncertainty between individual models regarding a change of stratospheric wind speeds is found at 60°N and 10 hPa (Manzini et al., 2014), the region where the strength of the SPV and SSWs are commonly diagnosed (Charlton and

45 Polvani, 2007). In line with this uncertainty, there is no agreement among CMIP5 and CMIP6 models on a possible change in frequency of SSW events (Ayarzagüena et al., 2018; Rao and Garfinkel, 2021). The multi-model mean suggests a slight increase in SSW frequency but the inter-model spread is large, even in the historical simulations (Rao and Garfinkel, 2021). Ayarzagüena et al. (2018) used 12 Chemistry Climate Model Initiative (CCMI) models for their analysis and found that most of them do not project a significant change in SSW frequency. Seviour et al. (2016) used two-dimensional diagnostics of the

50 SPV to differentiate between vortex splits and displacements and found that most CMIP5 models show some bias in simulating SSWs. Hall et al. (2021) made similar findings with CMIP6 models and found no notable improvement compared to CMIP5. They found that the chemistry schemes used in the models might influence the results. Other reasons for the large inter-model spread are assumed to be the different mean SPV strengths and differences in upward-propagating wave activity flux (Wu and Reichler, 2020).

The large inter-model spread and the uncertainties in the SPV response to climate change underline the need to investigate the reliability of climate models in simulating the SPV form, strength and stability. Previous studies mostly used a single run from each climate model. Single-model realizations limit analysis as to whether the differences between the models are caused by different model physics or by natural variability (Blanusa et al., 2023; Deser et al., 2020). Thus, particularly the high variability



of the wintertime NH stratosphere requires analysis using a large ensemble size (Deser et al., 2020). In the following, we aim

to assess how reliable recent large-ensemble model simulations are in representing the SPV and its spatial variability. For this, we compare in Sect. 3 geometric SPV diagnostics in large climate model ensembles with ERA5 reanalysis data using rank histograms (Matthewman et al., 2009; Seviour et al., 2013). Furthermore, the reliability of the ensembles in detecting SSWs separated into SPV splits and displacements will be analyzed in Sect. 4. Due to the large SPV variability and the high SSW frequency in the NH, we limit our analysis to the NH SPV. In Sect. 5 we discuss our results with regard to possible reasons for

the detected model differences and we finish the paper with some concluding remarks in Sect. 6.

## 2 Methods

### 2.1 Data

For this climate model assessment on SPV strength, form and stability, we use large climate model simulation ensembles. Each ensemble consists of multiple simulation members, which only differ by modified initial conditions, otherwise the model

physics and setups are identical (Deser et al., 2020). The advantage of an ensemble in comparison to a climate model with only one realization is that we can estimate the internal variability of a variable from the spread among the individual ensemble members. In our analysis, we use climate models from the Multi-Model Large Ensemble Archive (MMLEA) provided by the US CLIVAR (Climate and Ocean - Variability, Predictability, and Change) working group on large ensembles (Deser et al., 2020) as well as ensembles from the Coupled Model Intercomparison Project 6 (CMIP6, Eyring et al., 2016).

We use the historical simulations of these ensembles where all CMIP5- and CMIP6-class historical forcings are included. Information about the 11 climate model ensembles we used for our analysis can be found in table 1. The selection criteria for our model database were availability of at least 10 ensemble members and the geopotential height to calculate the SPV moment diagnostics (see section 2.2) at the pressure levels 10, 50 and 100 hPa. However, not all levels are available in all models. For reference, we compare the ensembles of the historical simulations with ERA5 reanalysis data (Hersbach et al., 2020). In

other words, ERA5 data serves as ground truth in our analysis. We use the same geopotential height-based SPV diagnostics for the reanalysis that we also apply to the model ensemble data.

The analysis is carried out for the period 1979–2014 covering years when as many observations as possible were assimilated in ERA5 including satellite observations (Hersbach et al., 2020). Vokhmyanin et al. (2023) found 22 SSWs in these 36 years of ERA5 data. We analyze daily data of the months from November through March in the NH, as this is when the SPV is usually

present and SSW disruptions happen.

### 2.2 Polar vortex moment diagnostics

To assess spatio-temporal SPV characteristics, the following 2-dimensional moment diagnostics are calculated: aspect ratio, kurtosis, centroid longitude and latitude, and objective area (for details of the calculation see Matthewman et al., 2009; Seviour



**Table 1.** Analyzed climate model ensembles from CMIP5 and CMIP6. Low-top and high-top models are separated by the horizontal line.

| Model | Ensemble members | Horizontal resolution (lat x lon) | Center of uppermost level in the vertical | Number of levels in the vertical | Reference |
|---|---|---|---|---|---|
| CanESM2[5] | 50 | 2.8°x2.8° | 1 hPa | 35 | Kirchmeier-Young et al. (2017)[5] |
| CanESM5 | 35 | 2.8°x2.8° | 1 hPa | 49 | Swart et al. (2019) |
| CESM2 | 49 | 0.9°x1.25° | 2.26 hPa | 32 | Danabasoglu et al. (2020) |
| CNRM-CM6-1 | 26 | 1.4°x1.4° | 86.4 km | 91 | Voldoire et al. (2019) |
| GFDL-CM3[5] | 20 | 2.0°x2.0° | 86.4 km | 48 | Donner et al. (2011) |
| INM-CM5-0 | 10 | 2.0°x1.5° | 0.2 hPa | 73 | Volodin and Gritsun (2018) |
| IPSL-CM6A-LR | 32 | 2.5°x1.25° | 80 km | 79 | Boucher et al. (2020) |
| MIROC6 | 10 | 1.4°x1.4° | 0.004 hPa | 81 | Tatebe et al. (2019) |
| MPI-ESM1-2-LR | 30 | 1.5°x1.5° | 0.01 hPa | 47 | Maher et al. (2019) |
| MPI-ESM1-2-HR | 10 | 0.4°x0.4° | 0.01 hPa | 95 | Müller et al. (2018) |
| UKESM1-0-LL | 16 | 1.25°x1.9° | 85 km | 85 | Sellar et al. (2019) |

[5] Model is part of CMIP5, other models are part of CMIP6. All models except UKESM1-0-LL and CNRM-CM6-1 do not have interactive chemistry. CNRM-CM6-1 has no typical interactive chemistry scheme, but a simplified interactive chemistry.

et al., 2013). In contrast to Matthewman et al. (2009) who described their method using potential vorticity, we will use the geopotential height to define the SPV edge, as suggested by Seviour et al. (2013). These metrics indicate whether the SPV is stretched, filamented as well as longitudinally and latitudinally displaced, and describe its area, respectively.

### 2.3 Rank histograms

We create rank histograms consisting of $n+1$ bins, where $n$ is the number of model ensemble members. For this, the values of the ensemble of a certain time step and variable are sorted in ascending order. For reference, the ERA5 reanalysis value at this time step is placed into this set at position $k$. The histogram counts of all bins greater or equal $k$ are then increased by one. This procedure is repeated for all available time steps (for details of the calculation see Hamill, 2001; Wilks, 2011). For a reliable (calibrated) ensemble, the counts should be uniformly distributed over all bins. If the ensemble deviates from the reanalysis, the shape of the histogram can be used to find out why (Wilks, 2011). For example, if the historical simulations are biased, there will be a linear trend in the histogram. When the counts of the bins are higher on the left and lower on the right of the histogram, the ensemble simulates the variable to be higher disproportionately often, which is called overforecasting bias. The opposite would be an underforecasting bias. If the ensemble under-/overestimates the variability, the ranks at the sides of the histogram have higher/lower counts than in the middle, which is called under-/overdispersion.

For objective assessment of the results, various measures can be considered, such as the $\chi^2$ statistic. This quantifies how close





the rank histogram is to an ideal uniform distribution. A perfectly flat histogram would produce a $\chi^2$ value of 0. Jolliffe and Primo (2008) introduced a method to split the $\chi^2$ statistic into multiple metrics where each one describes a particular shape of the histogram. The linear trend can be used as a bias indicator and the U-shape indicates over- or underdispersion. These metrics can be especially helpful when both bias and over- or underdispersion are present in an ensemble, as this can be difficult

to distinguish from the rank histogram alone.

Due to internal variability, it is possible that a rank histogram has a somewhat uneven distribution, even though the ensemble is covering the probabilities correctly. This can even result in an insignificant $\chi^2$. To determine which deviations from a uniform distribution can be attributed to internal variability, Suarez-Gutierrez et al. (2021) suggested the use of 'a perfect model range'.

To obtain this range a rank histogram is created for each ensemble member where this specific member is treated as reference (i.e., as if it was the reanalysis). This results in slightly different values for each bin in the rank histograms, depending on the member in question. The perfect model rank range is then defined by the range where 90 % (5th - 95th percentile) of the bin counts are found. Since a member from the ensemble can never be higher or lower than all ensemble members, the values for the rank range in the first and last bin are ignored.

**2.4  SSW diagnostics**

SSWs events can be subdivided into SPV splits and displacements and can be detected by means of the metrics described above. Seviour et al. (2013) suggested that an SPV split can be detected by using the aspect ratio and they defined values higher than 2.4 to be a split. For a displacement, the centroid latitude is arguably the best indicator. Here, Seviour et al. (2013) defined that a displacement has taken place if the centroid latitude is lower than 66$^o$N.

To assess how well the probability of these events is represented in the model simulation ensembles, the receiver operating characteristics (ROC) curves are used. The area under the ROC curve (AUC) indicates how well an ensemble is able to discriminate between SSW and non-SSW events using thresholds above (with reference to ERA5): It ranges from 0 to 1. A value of 1 and 0.5 indicates perfect skill and random guessing, respectively. To determine the uncertainty of the AUC, we provide error bars using the approach of the perfect model range, where we assume each ensemble member as observation.

We also reproduce the methodology from Hall et al. (2021) as previously applied in Mitchell et al. (2011) and Seviour et al. (2013) to examine relationships between modal centroid latitude and aspect ratio and displacement and split SSW frequency, respectively.

**3  Analysis of geometric polar vortex diagnostics**

In the following, the agreement between the ERA5 reanalysis and the historical simulations of the climate model ensembles

will be compared by means of the SPV moment diagnostics that were introduced in section 2.2. Rank histograms are discussed for all available models at 10, 50 and 100 hPa, however, the figures for 50 hPa and for 100 hPa are shown in the Supplement.





### 3.1 Aspect ratio

The aspect ratio is of particular interest for SPV dynamics because it is an indicator for vortex splitting events (Seviour
et al., 2013). Fig. 1 shows rank histograms of the aspect ratio for all analyzed climate model ensembles together with the
above explained statistical values $\chi^2$, bias and spread at 10 hPa, the level that is most commonly used to detect SPV splits.
As indicated in Sect. 2, interpretation of all results here are with reference to ERA5 reanalysis data, which is considered as
ground truth for the analysis. All models succeed to simulate the spread of the aspect ratio, but most models are biased to
some extent. At 10 hPa, four models are biased simulating lower aspect ratios more frequently than the reanalysis (CanESM2,
CanESM5, CESM2 and CNRM-CM6-1) and four models show a high bias (IPSL-CM6A-LR, MIROC6, and both MPI-ESM1-
2 ensembles). Only two models show no significant bias (INM-CM5-0, UKESM1-0-LL).

The strongest biases are found in the CanESM2 and CESM2 ensembles. Such strong biases can have implications for the
simulation of SPV splits. Frequent simulation of low aspect ratios points towards an underestimation of SPV split probability,
which will further be investigated in Section 4.2.

The rank histograms at 50 and 100 hPa can be found in the supplement (Figs. S1, S6). The models generally can be separated
into three groups according to their behavior compared to 10 hPa. One group of models shows larger biases at lower altitudes
(CanESM2, INM-CM5-0 and UKESM1-0-LL). In INM-CM5-0 and UKESM1-0-LL the low bias in aspect ratios appears only
at the two lower levels. In the other ensembles the bias is weaker at lower altitudes.

### 3.2 Centroid latitude

Fig. 2 shows the rank histograms of the centroid latitude for all analyzed climate model ensembles at 10 hPa. Most ensembles
show a bias in centroid latitude, but the spread is generally represented well. This is similar to the results of the aspect ratio.
The direction of the biases is not consistent among the models. The CanESM2, IPSL-CM6A-LR and MPI-ESM1-2 ensembles
simulate a high latitude bias with regard to the reanalysis, while the CESM2, INM-CM5-0 and MIROC6 ensembles show
a low latitude bias. Only CanESM5, CNRM-CM6-1 and UKESM1-0-LL do not show any notable bias or spread, i.e. the
160 corresponding statistical diagnostics show low values. Here, CanESM5 (see Fig. 2**B**) shows a notable improvement compared
to its earlier version CanESM2 (see Fig. 2**A**). The combination of biases towards higher centroid latitudes and lower aspect
ratios can only be seen in CESM2 (see Fig. 2**C**), which can explain the general underestimation of SSWs in this model (see
Sect. 4 and 5).

The rank histograms of the centroid latitude at the two lower pressure levels are shown in Figs. S2 and S7. In most models
the bias is similar at the different levels, which means that the performance of most ensembles with respect to centroid latitude
is not very sensitive to altitude. Only in CanESM5 a bias towards lower latitudes appears at the two lower levels. In MIROC6
the bias to simulate lower latitudes is only present at 10 and 50 hPa; at 100 hPa the bias vanishes completely and the histogram
is almost perfectly flat.

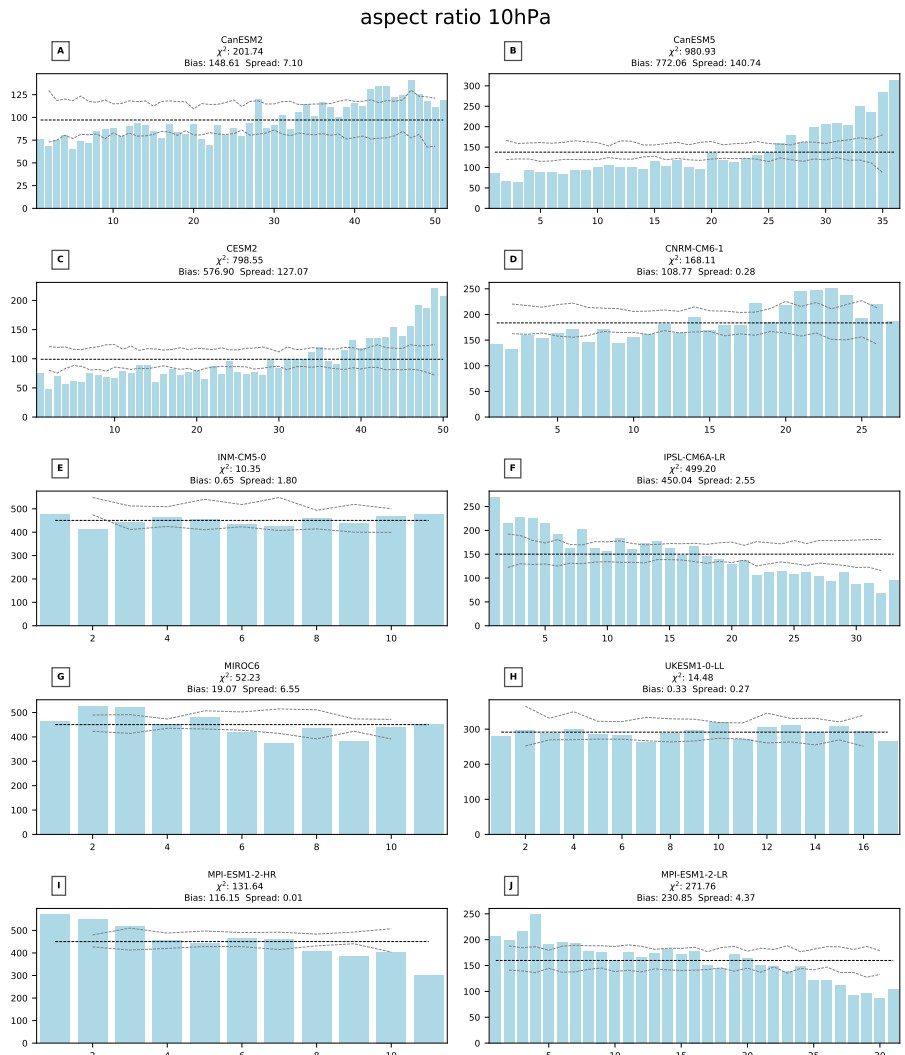

**Figure 1.** Rank histograms of aspect ratio at 10 hPa of all analyzed climate ensembles with their respective statistics (see Sect. 2). Blue bars show counts for the individual bins, the black dashed line corresponds to the expected value for a flat histogram, gray dashed lines indicate the perfect model range. The x-axis shows the ensemble member number and the y-axis shows the count of the bins.





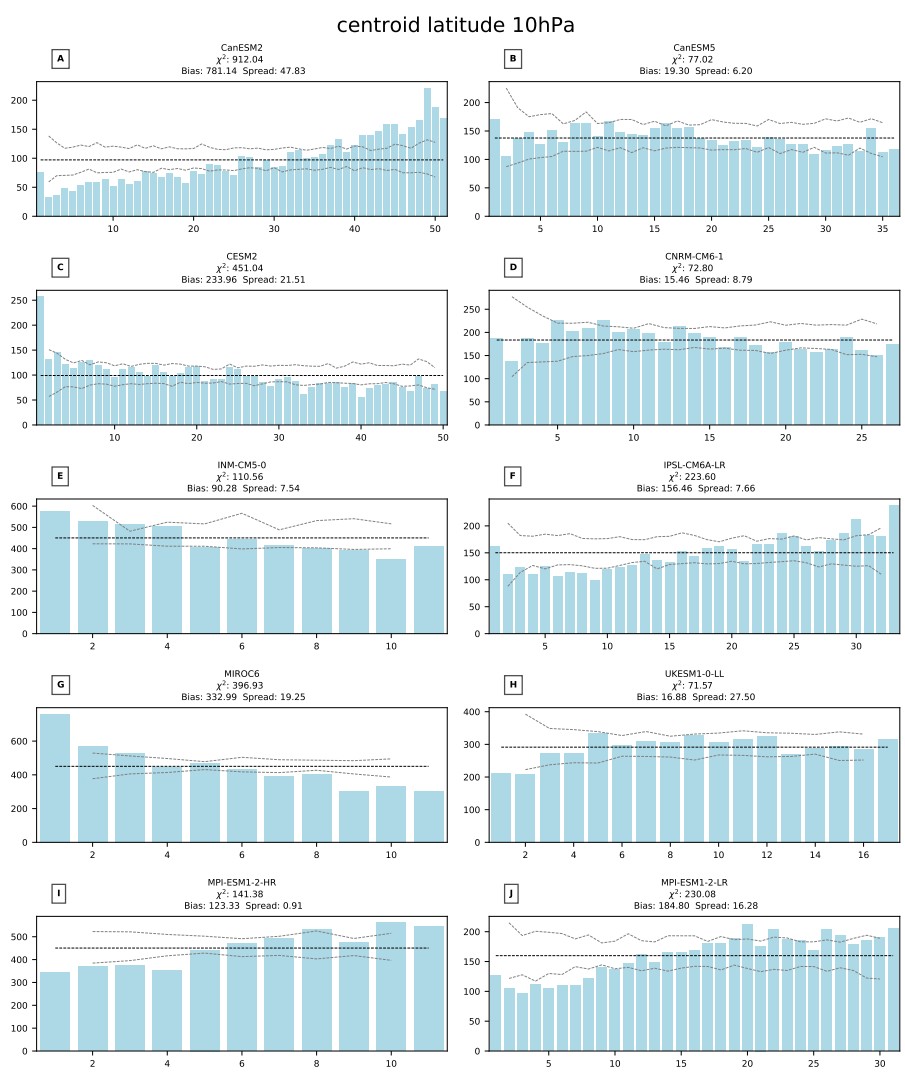

**Figure 2.** As Fig. 1, but for centroid latitude.



### 3.3 Centroid longitude

The centroid longitude in the climate models ranges from -180° to +180°, the negative values lie in the Western Hemisphere and the positive ones in the Eastern Hemisphere. The centroid longitude rank histograms (Fig. 3) show where the climate models over- or underestimate the position of the SPV. When the counts are lower/greater than average, the ensemble simulates the SPV center more/less frequently at the respective longitude. The centroid longitude is depicted best by the CNRM-CM6-1 (see Fig. 3**D**) and INM-CM5-0 (see Fig. 3**E**) ensembles. Other ensembles show notable deviations from a flat histogram, but there is no consistency in the deviations among the different models. The CanESM2, CESM2 and UKESM1-0-LL ensembles simulate the SPV center in the Eastern Hemisphere more frequently than the reanalysis. The IPSL-CM6A-LR and MIROC6 ensembles show the opposite bias. The rank histograms of CanESM5 (see Fig. 3**B**) and the MPI-ESMs (see Fig. 3**I-J**) show lower counts on both ends, indicating that the ensembles simulate the SPV more frequently in and around the region of the Bering Strait and Alaska (the meridian of +180/-180°) than the reanalysis. The rank histograms of the centroid longitude at 50 and 100 hPa are shown in Figs. S3 and S8. Some models show a change in the direction of bias or dispersion at the different pressure levels (e.g. CanESM2, IPSL-CM6A-LR, MIROC6). This could not be seen for the centroid latitude. In most models both bias and spread are present at least in some pressure levels. Only CNRM-CM6-1 produces a flat histogram where almost all counts lie inside the perfect model range in all three pressure levels. The UKESM1-0-LL and MPI-ESM1-2 ensembles show flat histograms at 50 and 100 hPa.

### 3.4 Kurtosis

The excess kurtosis is a measure for how the values of geopotential height are distributed within the SPV region (Matthewman et al., 2009). Mitchell et al. (2011) proposed that this diagnostic can be used to detect both SPV split and displacement events. They showed that exceptionally low values are often a sign that an SPV split has occurred, high positive values on the other hand can occur after splits and displacements (see their Figs. 2 and 5). The rank histograms for the excess kurtosis at 10 hPa are shown in Fig. 4. Four models show similar rank histograms with much higher counts on the left side of the histogram, namely CanESM2, CanESM5, CESM2 and CNRM-CM6-1. These ensembles underestimate the variability of the kurtosis and additionally simulate a kurtosis high bias. The result is that very low values of the kurtosis are simulated much less frequently in the models than they occur in the reanalysis. Therefore, these models likely underestimate the SPV split frequency. This is in line with the aspect ratio low bias in these models (see Sect. 3.1), except for CNRM-CM6-1 (see Fig. 4**D**). The UKESM1-0-LL (see Fig. 4**H**) ensemble shows a similar kurtosis behavior as the four model ensembles from above, but the deviations are by far not as pronounced. Even though the UKESM1-0-LL ensemble shows a good representation of the aspect ratio, it simulates low values of the kurtosis less frequently. This is in line with the results by (Hall et al., 2021) that the model simulates too few split events. The MIROC6 and INM-CM5-0 ensembles perform best in representing the kurtosis, in particular at 10 hPa. Despite the differences in vertical and horizontal resolution, there is no substantial difference between the two MPI-ESM1-2 simulations. However, we conclude that if the spatial model resolution has a substantial influence on the representation of the SPV, dedicated experiments with systematic resolution changes will be needed in future studies. At the two lower altitudes





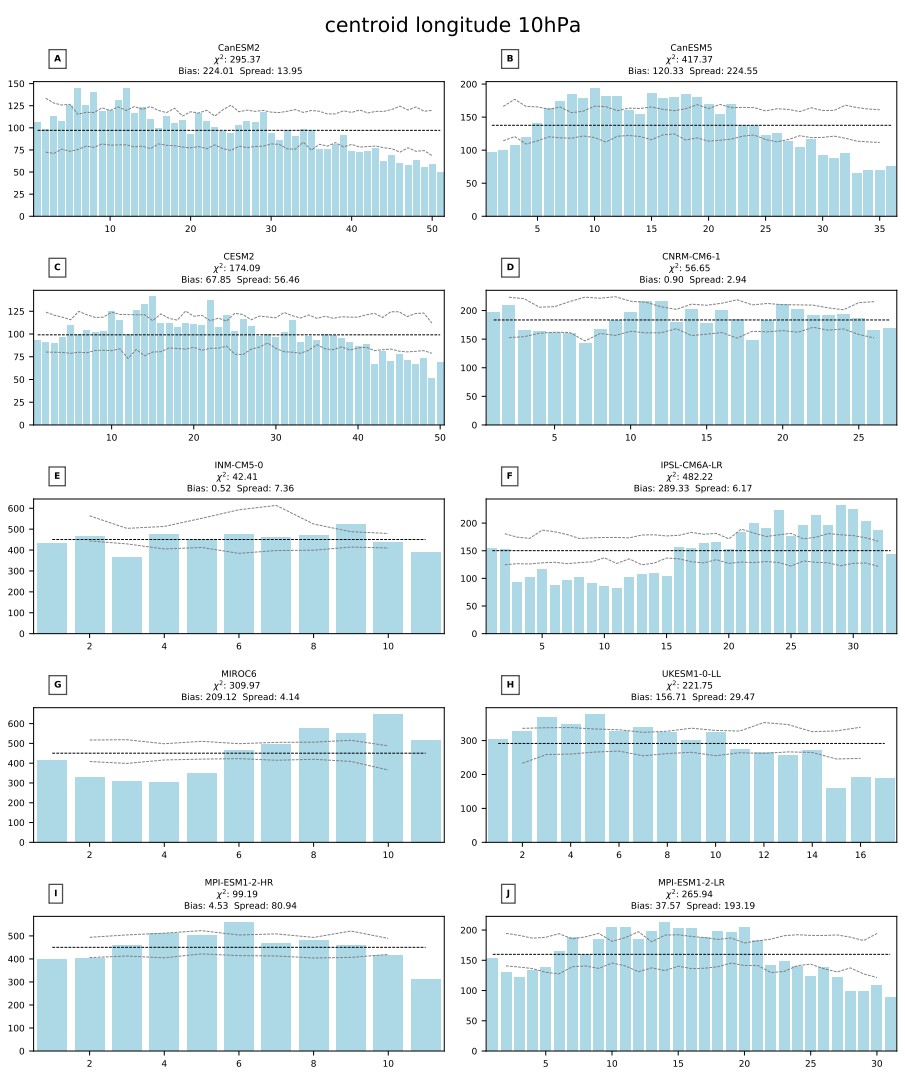

**Figure 3.** As Fig. 1, but for centroid longitude.



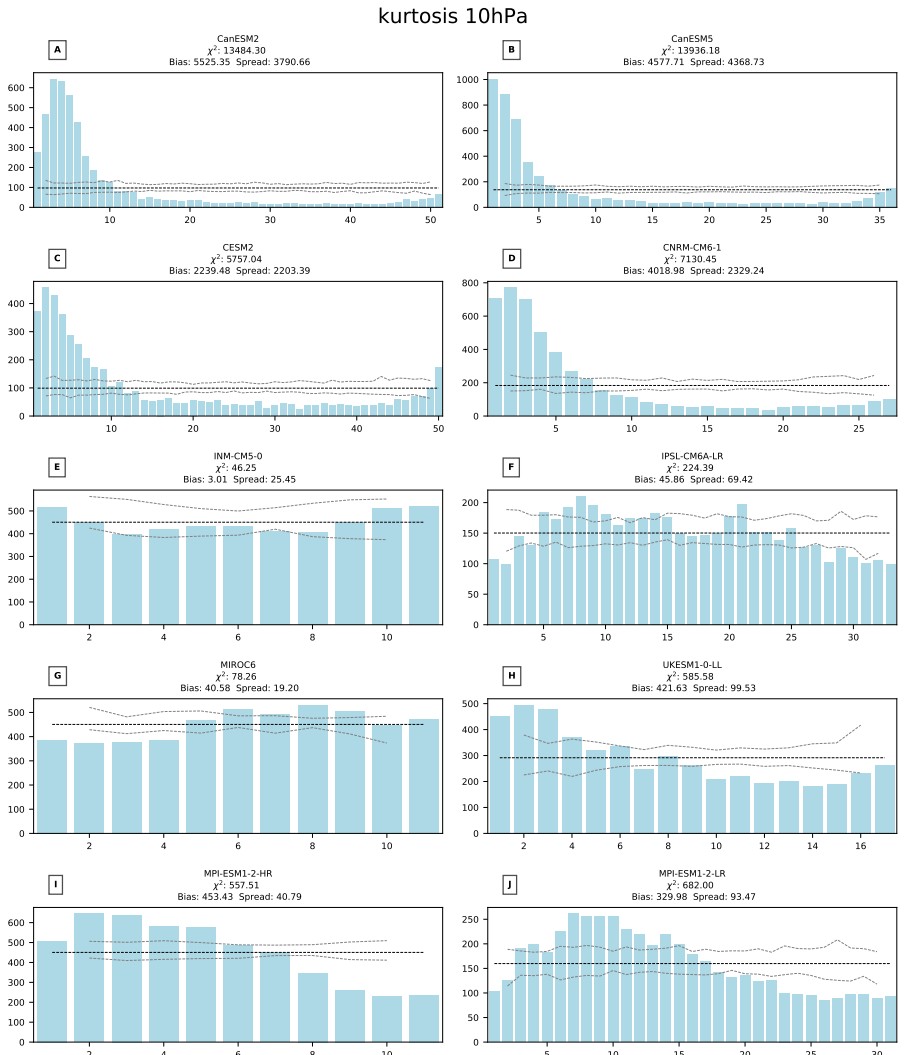

**Figure 4.** As Fig. 1, but for kurtosis.

(Figs. S4 and S9), most ensembles underestimate the kurtosis variability (CanESM2, CanESM5, CESM2, INM-CM5-0 and UKESM1-0-LL and GFDL-CM3 at 100 hPa). Only the IPSL-CM6A-LR and both MPI-ESM1-2 ensembles overestimate it. The MPI-ESM1-2 ensembles show almost flat rank histograms at 100 hPa. Generally, in comparison with centroid latitude or

205 aspect ratio most models do not simulate the kurtosis well with respect to ERA-5, in particular at 10 hPa. This suggests that centroid latitude and aspect ratio seem to be more reliable indicators for SPV split and displacement frequency estimates than the kurtosis.



## 3.5 Objective area

The objective area is of interest because a larger/smaller area of low geopotential height is often related to a stronger/weaker
SPV with higher/lower wind speeds. Multiple ensembles (INM-CM5-0, IPSL-CM6A-LR, MIROC6 and both high and low
resolution MPI-ESM1-2 ensembles) simulate a strong small SPV bias at 10 hPa (Fig. 5). In addition to that, these models un-
derestimate the variability of the objective area. This combination results in a strong underestimation of a large SPV occurence.
The CanESM2 (see Fig. 5**A**) ensemble also shows this combination, but not as pronounced as the above-mentioned models.
The CanESM5, CESM2, CNRM-CM6-1 and UKESM1-0-LL ensembles simulate a large SPV bias, which is likely connected
to too high wind speeds.

At the lower altitudes (see Figs. S5 and S10) most models are biased in the same direction as at 10 hPa, but in some models
the strength of the bias varies with height. Only INM-CM5-0 shows a small SPV bias at 10 hPa (see Fig. 5**F**) and a large
SPV bias at 50 hPa. At 100 hPa barely any bias can be detected in INM-CM5-0. Models with a large SPV bias at 100 hPa,
also simulate a low aspect ratio bias at 10 hPa (CanESM2, CanESM5, CESM2) and vice versa (IPSL-CM6A-LR and both
MPI-ESM1-2 ensembles - except for MIROC6). Models without any notable bias (irrespective of the spread) for the objective
area show a good representation of the aspect ratio (CNRM-CM6-1, INM-CM5-0, UKESM1-0-LL). A connection between
lower stratospheric winds and SPV split frequency in the CMIP6 models was already noted by Hall et al. (2021). They found
that the frequency of SPV splits was related to the wind speeds at 100 hPa, because higher wind speeds hinder the upward
propagation of wave number 2 planetary waves into the stratosphere. Similarly, Wu and Reichler (2020) showed that the
highest uncertainty in the frequency of SSWs comes from an uncertainty in lower stratospheric wind speeds. Most models
do not succeed to simulate the objective area of the SPV adequately. Most commonly, they underestimate the variability and
additionally a bias is often present. The best representation at 10 hPa is shown by the UKESM1-0-LL ensemble. For the two
lower levels, CNRM-CM6-1 is the closest to a perfect representation, confirmed by the lowest values of the $\chi^2$-statistic.

## 4 Sudden stratospheric warming analysis

When studying the NH SPV, a particular focus lies on events when the SPV is disrupted, so called sudden stratospheric
warmings (SSWs). We here assess the ability of the climate models to distinguish between SSWs and steady SPV conditions.
While the rank histograms reveal reliability (consistency), they do not evaluate statistical resolution (the degree to which a
forecast sorts the observed events into different groups), so our study needs to be accompanied with other tools such as the
ROC (Hamill, 2001). Using the ROC we can analyze how well the different models are able to simulate SSW events through
diagnostics of the SPV morphology. The applied method allows us to individually diagnose displacement- and split-type
SSWs. Hence we conduct two separate analyses here, as it has been shown that these events, as well as their surface impact,
fundamentally differ (Baldwin et al., 2021, and references therein).





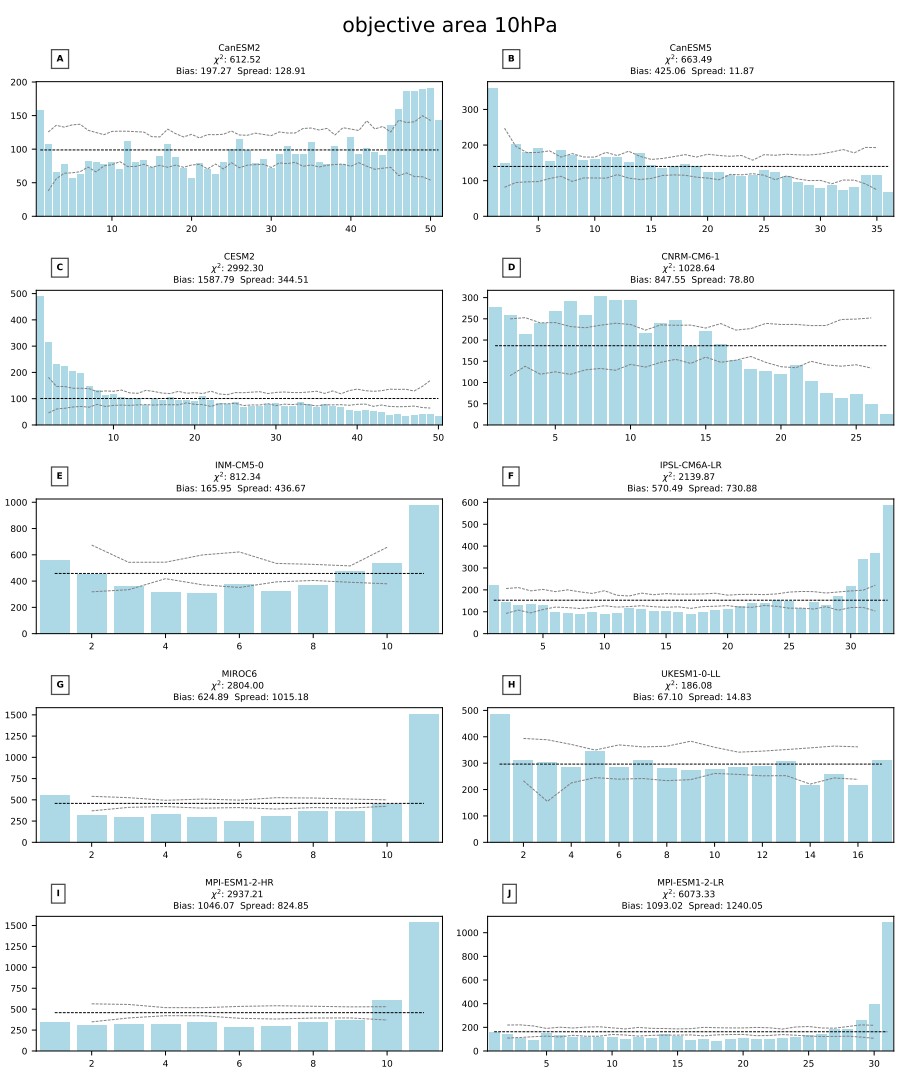

**Figure 5.** As Figure 1, but for objective area.





## 4.1 Displacement events

Fig. 6 shows the areas under the ROC curves (AUC) of the analyzed climate models for detection of SPV splits and dis-
240 placements. All models reach a value of greater than 0.5 which means their performance is better than randomly guessing. As
hypothesized (Hall et al., 2021), SPV displacements (see Fig. 6**A**) may be less frequent in models that simulate a high centroid
latitude bias. However, the ROC diagram is insensitive to certain types of biases (Kharin and Zwiers, 2003), since a biased
model may still have good statistical resolution. However, the ROC can still be considered as a potential skill when the model
is correctly calibrated (Wilks, 2011). This fact can also explain the differences between CanESM2 and CanESM5. According
to the AUC, CanESM2 distinguishes between displacement events and stable SPV conditions better than CanESM5. This is re-
markable as CanESM5 was performing much better in the rank histogram analysis for the centroid latitude at 10 hPa compared
to CanESM2.

All low-top models (see Tab. 1) generally reveal lower AUC values than the high-top models with the exception of MPI-
ESM1-2-HR. In fact, MPI-ESM1-2-HR has the lowest value of all analyzed models. Additionally, the AUC for the MPI-ESM1-
250 2-HR ensemble is slightly lower than for its low resolution counterpart: MPI-ESM1-2-LR. The CNRM-CM6-1 ensemble shows
the best performance regarding the simulation of SPV displacement events. This model also has one of the best representations
of the centroid latitude at 10 hPa in the rank histograms. In fact, the rank histogram was similar to CanESM5 for which a rather
weak performance in the ROC curves was found. A reasonable performance is shown by the INM-CM5-0 ensemble, which
in fact has the second highest AUC. IPSL-CM6A-LR, MIROC6 and MPI-ESM1-2-LR show similar values for the AUC. The
255 UKESM1-0-LL ensemble shows an average performance compared to the other climate models. Again, this stands in contrast
to the rank histograms where this ensemble was closest to a flat histogram with the lowest $\chi^2$ statistic of all models.

These results demonstrate that even if the rank histogram implies a good representation, i.e. a reliable ensemble, it can show
a comparatively low statistical resolution in distinguishing between displacements and non-displacements (e.g CanESM5).
Generally, for most of the ensembles the AUC lies in a narrow region around 0.6, implying that the simulation of displacement
events can still be improved in the climate ensembles, e.g. by calibration. It has been suggested by Seviour et al. (2016) and
Hall et al. (2021) that models with a bias in centroid latitude also have a bias in displacement frequency in the respective
direction. However, Fig. S11 shows that we could not reproduce a clear relationship between number of displacements and
modal centroid latitude from Fig. 3a in Hall et al. (2021). As centroid latitude diagnoses the climatological mean position,
and the displacements particular extreme events of this position, it is reasonable that there is no clear relation, although the
265 diagnostics are connected.

## 4.2 Split events

In general, the climate model ensembles do not simulate SPV splits (see Fig. 6**B**) as well as SPV displacements (see Fig. 6**A**)
In fact all models except for IPSL-CM6A-LR have lower AUCs for split events than for displacements. Overall, a weaker
performance of the low-top models can be detected, which is particularly obvious for CanESM5. As in the displacement
analysis, CanESM2 performs better than its newer counterpart CanESM5. In the latter the AUC is even lower than 0.5 meaning



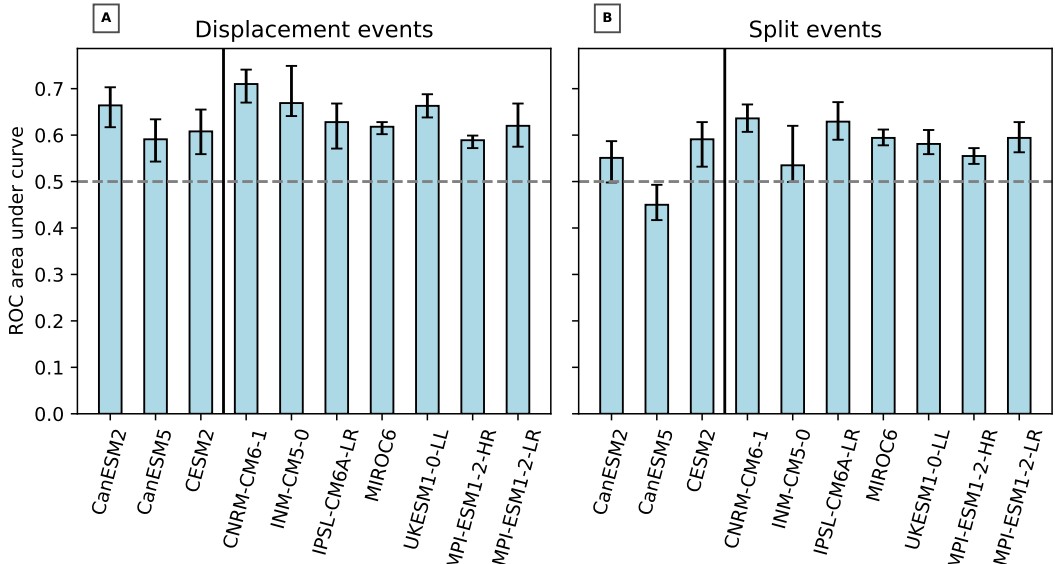

**Figure 6.** Area under the curve for the ROC curves of the analyzed climate ensembles for displacement (left) and split (right) events. Grey line lies at 0.5, the value at which the simulation is not better than randomly guessing. Low-top models are separated on the left with a black line. Error bars indicate 5th and 95th percentile estimated by using the perfect model range.

that in CanESM5 the false positive rate for detecting split events is higher than the true positive rate. In fact, CanESM5 is the only model that produces an AUC of lower than 0.5 for split events, even when considering the error bars. A reason for this might be the strong bias to lower aspect ratio at 10 hPa that is likely resulting in underestimation of the frequency of SPV splits, which are associated with exceptionally high aspect ratios. The CESM2 ensemble shows a similarly strong bias compared to CanESM5 but has a better representation of split events according to the ROC plots.

CNRM-CM6-1 reaches the largest AUC as was already seen for the displacement events. Thus, this model can be regarded to have the best representation of SPV displacements as well as splits and likely SSW events in general. The AUC of the INM-CM5-0 ensemble reaches a value of slightly above 0.5, indicating that the simulation of splits is only marginally better than randomly guessing their occurrence. This result stands in contrast to the fact that this ensemble has shown one of the best

performances in the rank histogram analysis for the aspect ratio without any significant bias or spread. This again corresponds to the insensitivity of the ROC to certain biases as discussed above. The IPSL-CM6A-LR ensemble on the other hand almost reaches the performance of CNRM-CM6-1 in spite of its bias to predict higher aspect ratio more often. MIROC6, MPI-ESM1-2-LR and UKESM1-0-LL show similar AUC that lie between the best and weakest performing ensembles. The high resolution MPI-ESM1-2 ensemble is showing a lower AUC than the low resolution version as it was already seen for displacement events,

but it still remains well above the value of 0.5.



Similar to displacement events, Seviour et al. (2016) and Hall et al. (2021) found that models showing a strong bias to lower aspect ratios in our analysis indeed underestimate the SPV split frequency. Fig. S12 reveals that the linear relationship between number of splits and modal aspect ratio from Fig. 3b in Hall et al. (2021) cannot be reproduced in large ensemble simulations of high-top models, only when low-top models are included. This underlines our results that good performance in representing the geometric-based diagnostics in rank histograms is not necessarily connected to a good performance in simulating displacements and splits. Wu and Reichler (2020) also demonstrated that bias-corrected models for vortex strength may not consistently align with reanalyses in terms of revealing SSW frequency.

## 5 Discussion and Outlook

We assessed the SPV in large CMIP5 and CMIP6 climate model ensembles using rank histograms with reference to ERA5 reanalysis data. The performance of the models varies depending on the analyzed variables and pressure levels. No model ensemble can be highlighted as having the best or worst performance over all variables and pressure levels. If the general performance over all levels and variables is regarded, the CNRM-CM6-1 and UKESM1-0-LL ensembles can be considered to be representing SPV form and variability best. These two models produce a flat rank histogram for most of the geometric variables at most altitudes, which means that the simulated SPV in these models agrees well with that of ERA5. The flat rank histrogram is a necessary but not a sufficient condition for concluding reliability in SPV simulation.

Furthermore, we used the ROC analysis in order to assess the ability of the ensembles regarding SPV displacement and split frequencies. As all models reach an area under the ROC curve of more than 0.5 (see Fig. 6**A**), they distinguish between SPV displacements and non-displacements better than random guessing. The ensembles represent displacement events better than split events in general. The best representation of both SPV splits and displacements has CNRM-CM6-1. This model performs well in the rank histogram analysis as well. However, a general rule of thumb that connects the rank histogram with the ROC analyses could not be found here. This is due to the insensitivity of the ROC to biases in the forecast. The ROC diagram can be considered as a measure of potential usefulness when a model ensemble is correctly calibrated. A joint analysis of variety diagnostics provides the bigger picture about the quality of large-ensemble model simulations.

The model top height seems to be a key factor. Low-top models reveal strong biases for most variables, in particular CESM2. Low-top models were already found to simulate fewer SSWs than in the reanalysis and a lower variability of the SPV wind speeds (Charlton-Perez et al., 2013; Hall et al., 2021). A better vertical resolution also improves the simulation of the SSW frequency (Wu and Reichler, 2020). The models with the finest spatial resolution (INM-CM5-0 and MIROC6) showed a good performance, especially for the aspect ratio and kurtosis at 10 hPa. Overall, this means that the downward influence of the upper stratosphere and mesosphere has a large influence on the SPV and on SSWs (Hitchcock and Simpson, 2014). This is not surprising, as large amounts of wave drag are deposited at high altitudes, which strongly influences middle atmosphere dynamics. In the low-top models, this influence is not adequately represented. Models with more levels in the vertical in the stratosphere generally perform better. CNRM-CM6-1, which has the second-highest number of levels in the vertical, does not only have a good representation of most variables but also the best results in detecting splits and displacements. The MPI-



ESM1-2 ensembles on the other hand produce similar results even though they differ in vertical and horizontal resolution. Dedicated model experiments with simulations in various horizontal and vertical resolutions are needed to systematically assess the impact of resolution on SPV representation.

An additional source of uncertainty might be the gravity wave (GW) parameterizations (e.g. Eichinger et al., 2020; Karami et al., 2022; Eichinger et al., 2023). Events with strong gravity wave drag can affect the refractive index in the lower strato-

sphere (Kuchar et al., 2022). A higher refractive index results in stronger upward propagating wave activity and thus the SPV is disrupted more easily. Wu and Reichler (2020) found that the uncertainty of the refractive index in the lower stratosphere above the tropospheric jet plays an important role for the uncertainty in the simulated SSW frequency. Therefore, different gravity wave parameterizations in the models may be attributed to these uncertainties (Sigmond and Shepherd, 2014).

Apart from model resolution and GW parameterizations and their tuning parameters, Morgenstern et al. (2022) revis-

ited stratospheric ozone chemistry influence on the SPV and SSW frequency. Several additional studies demonstrated the importance of interactive ozone chemistry for representing temperature variability and extremes in the Arctic polar strato-sphere (Haase and Matthes, 2019; Rieder et al., 2019; Oehrlein et al., 2020). Therefore, the way how atmospheric chemistry is treated in the model may be another factor for model skill in representing the SPV, in particular the feedback of stratospheric ozone on dynamics via radiation. CNRM-CM6-1 has a simplified but still interactive chemistry (Voldoire et al., 2019). The

only analyzed model with a complete interactive chemistry is UKESM1-0-LL and overall it performs well. However, a detailed analysis of its impact on spatio-temporal SPV variability would be needed for conclusive statements.

Models that were found to simulate well the alternating easterlies and westerlies in the tropics by Richter et al. (2020) (the quasi-biennial oscillation, QBO), mostly perform better in our analysis (e.g., CNRM-CM6-1, IPSL-CM6A-LR, UKESM1-0-LL). On the other hand, models with poor QBO representation (CanESM2, CanESM5, CESM2) show a weaker performance

in the rank histograms and the representation of splits and displacements. The SPV is influenced via teleconnection associated with the QBO, via the so-called Holton-Tan mechanism (HTM; Holton and Tan, 1980; Baldwin et al., 2001). Rao et al. (2020) analyzed which models have a good representation of the HTM, but here we find no clear connection of a good HTM representation with a good representation of SPV variability.

While a relatively long period was regarded in the rank histogram analysis, it cannot be ruled out that an ensemble might

show different performances during this time (Bothe et al., 2013). An option could be to analyze individual months separately, since differences in the model performance might for example occur between mid-winter, where the highest variability in wind speeds is observed, and early as well as late winter.

As has been investigated before (Hamill, 2001; Bröcker, 2008; Siegert et al., 2012), a flat rank histogram is only a nec-essary but not a sufficient criterion to conclude that ensemble simulations are reliable (calibrated). Due to the unknown and

undersampled nature of initial-condition distributions and unavoidable simplifications and errors in the dynamical formulation, we cannot expect raw ensembles to be calibrated (Wilks, 2019). This can be achieved via calibration (Vannitsem et al., 2018; Wilks, 2018) which can likely reduce uncertainty in climate projections (Tett et al., 2022).

Furthermore, the thresholds used for the definition of the events could be varied. Other values might lead to better resolution between steady and unsteady SPV conditions. The thresholds we used here were chosen based on the reanalysis dataset by



Seviour et al. (2013) as stated in Sec. 2.4. Another important question is whether the number of ensemble members is sufficient for evaluation of the highly variable SPV representation. In particular, the INM-CM5-0, MIROC6 and MPI-ESM1-2-HR ensembles may not be large enough to fully cover effective dimension of SPV (Christiansen, 2021). This is a topic for detailed future investigation.

## 6 Conclusions

In this study, we assess the stratospheric polar vortex (SPV) form, stability and variability as well as the ability to distinguish different morphologies of sudden stratospheric warmings (SSWs) in large CMIP5 and CMIP6 climate model ensembles. We analyze the SPV by means of rank histograms and the SSWs separated into splits and displacements by receiver operating characteristics curves and use ERA5 reanalysis data as reference. These analyses reveal strongly varying performances of the individual models over all SPV moment diagnostics and pressure levels. The two models that overall simulate the SPV and

SSWs closest to ERA5 are CNRM-CM6-1 and UKESM1-0-LL. In general, the ensembles show a better ability in simulating displacement-type SSW events than split-type events. As SSWs represent extreme events, this model skill, however, is not always connected with representing well the geometry-based SPV diagnostics centroid latitude and aspect ratio, which diagnose SPV climatologies.

For the SPV centroid latitude and aspect ratio most ensembles are biased to some extent but with no consistent direction

among the ensembles. Our regression of these biases did not indicate robust biases in split and displacement frequency as in Seviour et al. (2016) and Hall et al. (2021). Of all analyzed diagnostics the kurtosis appears to be the hardest to simulate correctly. Most of the ensembles underestimate the variability of the kurtosis. Strong biases and an underestimation of the variablity is found for the SPV area as well. Overall, this may be constituted by the difficulty of models to simulate the well-known non-linearity of stratospheric dynamics (Matthewman and Esler, 2011; Cohen et al., 2014; **?**) and calls for caution when

using these diagnostics as SSW proxies.

We can conclude that models with a higher lid and models with a finer vertical resolution generally simulate the SPV and SSWs better with reference to ERA5. However, many factors influence the SPV properties and the SSW frequency, such as interactive chemistry, gravity wave parameterizations and other dynamical processes that differ in the individual models. It is therefore not clearly assignable from this study which model characteristics are the decisive ones for representing well the SPV

and its behaviour. However, our findings set the basis for sensitivity experiments with individual models changing individual model components, such as vertical/horizontal resolution, gravity wave scheme, chemistry etc.

Knowledge of how well different climate models perform in simulating the SPV and SSWs correctly is of utmost importance for finding the adjustment screws to improve their performance, as well as for assessing their reliability in future climate projections. The latter is particularly important with regard to polar stratospheric ozone and its evolution across the 21st

century.



*Code availability.* The code that was used to produce all plots in this study is available via Zenodo (Kuchar and Öhlert, 2023).

*Data availability.* All processed data files for this study are provided via Mendeley Data (Kuchar, 2023).

*Author contributions.* AK designed the study. AK and MÖ analysed the data. MÖ and AK compiled the manuscript with inputs of all other authors.

*Competing interests.* The authors declare that they have no conflict of interest.

*Acknowledgements.* This study has been supported by Deutsche Forschungsgemeinschaft under grant JA836/43-1 and within the Transregional Collaborative Research Centre SFB/TRR 172 (Project-ID 268020496), subproject D01. RE acknowledges support by the Czech Science Foundation (grant no. 21-03295S). Climate model output has been kindly provided by the US CLIVAR Working Group on Large Ensembles and by the Earth System Grid Federation (ESGF). We acknowledge the World Climate Research Programme, which, through its
Working Group on Coupled Modelling, coordinated and promoted CMIP6. We thank the climate modeling groups for producing and making available their model output, the ESGF for archiving the data and providing access, and the multiple funding agencies who support CMIP6 and ESGF. For data processing, resources have been used at the Deutsches Klimarechenzentrum (DKRZ) under project ID bd1022.



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
