# Peer review of "Large ensemble assessment of the Arctic stratospheric polar vortex morphology and disruptions"

_EGUsphere, 2023_

## Referee Comment (RC2)

**GENERAL COMMENTS**

The manuscript assess the representation of the geometry of the winter SPV in a few model ensembles, and compares the model ensembles' SSW and SPV frequency to the observed. To persue this analysis, geopotential height data over different stratospheric levels and at daily frequency is taken from the historical simulations of climate models participating in CMIP5 and CMIP6. Based on their results, the authors filter the models that are best at representing the SPV spatial distribution and SSW frequency, and conclude that a good calibration of the SPV shape does not necessarily correspond with an adequate representation of split and displacement SSWs. Outlook on the possible reasons underlying inter-model differences is also provided.

Although the topic is of interest for the community and the results might be useful for the institutions developing the different models, I find that the present manuscript is of poor quality. This is because (i) the text is fragmented and difficult to follow, (ii) the language is often imprecise, (iii) the methods are not clearly presented, (iv) the many differences across models and resolutions are difficult to grasp. The whole manuscript needs to be generally and carefully revised before being fit for publication in WCD. In the following, some suggestions that should help improve the afore-mentioned issues are included.

**SPECIFIC COMMENTS**

1. To improve the text please work on fluency and on the precision of the wording. Often adjacent sentences repeat the same concepts, or are fragmented, or badly connected.  For example, see some of the points in the TECHNICAL CORRECTIONS.

2. The methods need to be expanded and/or clarified. Specifically, the meaning of the geometric diagnostics can be better explained, the methods behind the rank histogram and ROC/AUC calculations are not clear.

3. It would be useful to compare the list of ERA5 SSW split and displacement events determined with geometric methods (i.e., those identified here) to those identified with more traditional methods (lists of events and their types are available online or in literature). Additionally, please provide the magnitude of the mean SSW frequency for each model, together with the AUC.

4. The Discussion and Outlook section is dispersive, and should be organised more clearly, without mixing your results (discussion and limitations) with the outlooks. Probably, some paragraphs in the Results section would fit better in the Discussion section (e.g., see TECHNICAL CORRECTIONS). Finally, it would be useful to add a table summarising the performances/biases of each model, by geometric diagnosis and by type of SSW event.

5. How does the map of the mean gh10 climatological bias look like for each model ? Is it relatable to the more specific rank histogram results ? If so, such a figure could be included at the beginning of the Results' section.

6. It would be useful to show an example of a ROC curve for one model, displaying both the split and displacement curve. This could go in the Methods or in the Results, and would be useful for explaining how the ROC curve is constructed.

7. Since you mention the diagnostics of the vertical structure of the SPV in the abstract, it is advisable to show a couple of meaningful plots (or statistics values) for the other stratospheric levels in the main manuscript.

8. You should mention any previous use and results of geometric diagnostics in your Introduction section. Or did I miss this ? On the other hand, SPV projections are not your main focus, so you should mention them briefly to motivate your work or more extensively only if directly related.

**TECHNICAL CORRECTIONS**

**General**

Please check the usage of the adjectives 'low'/'high' (e.g. related to bias), and replace with 'weak'/'strong' or 'negative'/'positive' when ambiguous.

**Abstract**

Line 1 : « each hemisphere ».

Line 1 : The word « phenomenon » is not adequate for describing the SPV. Please change.

Lines 5-13 : Abstract would gain in clarity if the description was separated into geometric diagnosis and split-displacement diagnosis (e.g. line 6 shifted after the geometric diagnosis). The fluency of the text would improve.

**Introduction**

Line 25 : « relate ».

Lines 25-26 : The term « windiness » is ambiguous. I would suggest to replace the first two elements in the list with « the position of the jet and the precipitation patterns over Europe and the Mediterranean region ».

Line 27 : « **where** the SPV is highly variable ».

Line 31 : « from **a** final warming».

Lines 32-33 : The sentence is difficult to read, could you please reformulate ?

Lines 37-39 : The sentence is out of place in the present position. Please fit adequately in the discussion of literature.

Lines 57-58 : Do you mean that a limitation of the single-member analysis is that one can't distinguish between natural and inter-model variability ? This is not the meaning emerging from the sentence, please correct or clarify.

Line 63 : «  is », for consistency of the verb tense.

Lines 63-63 : The following sentence/content should be shifted in an adequate position, before the outline. « Due to the large SPV variability and the high SSW frequency in the NH, we limit our analysis to the NH SPV. »

**Methods**

Lines 70-72 : You can remove the reasoning here, as it is already discussed in the Introduction.

Line 76 : Please change to « climate models used **in** our analysis **are described** in table 1 ».

Line 78 : Inconsistency between the two sentences. Please modify.

Line 80 : The first part of the line is accessory, please remove or include in compact form as part of the previous sentence.

Line 91-92 : The last sentence gives a qualitative description of the different moment diagnostics, but the link with the diagnostics (two sentences back) is not obvious – I suggest giving this information just following the list of diagnostics. Moreover, an extension of their description, here or at the beginning of each section in the results, would be welcome for any non-expert reader.

Lines 95-99 : The description of rank histograms is confusing. Please describe precisely and clearly how the counts within each histogram are determined, and why the distribution should be uniform in a well-calibrated forecast. Is the sentence « The histogram counts of all bins greater or equal k are then increased by one » correct? From what I read from Sect. 2 in Hamill et al. 2001 (your ref.), only one of the bins is updated at each timestep, corresponding to the forecast ensemble bin where the observation falls.

Line 119 : Mention where and how the perfect model range method is used within the present work.

Line 122-24 : It is not clear to me if your definition of displacements and splits corresponds to the description you give in these lines. Could you make it more explicit ?

Line 126 : Insert ref. for ROC and AUC.

Line 127 : Can you describe in this paragraph how the thresholds for drawing the ROC curve are computed ? And what you mean by true and false positive rates in a historical simulation (rather than in a forecast) ? An example of a ROC curve would also be welcome, including descriptive labels for x and y axes.

**Results**
Line 139 : It would be useful to specify again, within this sentence, that high values of the aspect ratio correspond to split events.
Line 143 : No need to repeat the aside « which is considered as ground truth for the analysis ».
Line 144 : «  simulate ».
Line 145 : «  positive ».
Line 146 : How is the significance of the bias computed ?
Lines 148-49 : You could merge the last two sentences of the paragraph, there is a partial overlap.
Line 152 : Does « low » mean 'weak' or 'negative' ?
Line 153 : « model ensembles ».
Line 165 : Please specify pressure levels each time.
Line 192 : « high » means positive ?
Lines 200-01 : I don't understand the sentence starting by « However.. ». Please clarify or remove.
Lines 212-14 : Please specify large objective area or strong SPV, or describe what large SPV means. Check the same in the rest of the subsection.
Lines 221-228 : These lines are better fitted for the Discussion section.
Line 240 : Specify that the assertion is valid for displacements. Or else present only the displacement results in the first introductory sentence.
Lines 242-44 : I find the discussion quite vague. Could you clarify ?
Sections 4.1 and 4.2 : The second part of each subsection is better fitted for the Discussion section, specially when the manuscript's results are compared with previous literature.

**Discussion and Outlook**
Lines 314-17 : These lines are rather vague and un-related from the results and conclusions of the manuscript. Please remove if not otherwise justified.
Line 323 : It would be good to acknowledge that you are not analysing or describing GW parametrisation in this work.
Line 328 : Is this phrased correctly ? If so, it is difficult to understand what you are concluding.
Lines 348-352 : This discussion is too general – it should connect more with the results of your work if you want to include it.

**Conclusions**
Line 360 : What do you mean by « stability » ?
Line 370-71 : Can you tell why your results are different from previous literature ? Is it a different metrics ?
Line 375 : At some point you had some contradictory results regarding resolution and SPV representation (see e.g. Line 199). Is it worth taking this into account in the present assertion ?
Line 380 : « variability » instead of « behaviour ».
Line 380 : Replace « set the basis » with « can constitute/be a reference ».
Line 380 : Rephrase « with individual models changing individual model components ».
Line 382 : « the SPV spatial variability ».
Line 383 : Please rephrase « for finding the adjustment screws to improve their performance ».

**Figures**
Figure 1-5 : Increase font size of model name and statistics above each plot. Mention the different models and the respective letter labels in the caption.

---

## Author Comment (AC3)

**Replies to reviews for:**
**Large ensemble assessment of the Arctic stratospheric polar vortex**

WCD, Ales Kuchar et al.

**1 Reply to reviews**

Dear Daniela Domeisen,

thank you very much for helping in the publishing process of this paper. We appreciate the helpful comments of the two
5 anonymous reviewers and will revise the manuscript accordingly, such that we hope it can then be published in WCD. Please
find our point-by-point answers to their comments below.

Best wishes
Ales, Roland, Maurice and Christoph

**2 Ref #1**

The authors present a mostly statistical assessment of CMIP model performance in relation to the geometry of the perturbed
polar vortex, plus the frequency of split or displacement SSWs. Their main tool of analysis are rank histograms created with
models from the Multi-Model Large Ensemble Archive against ERA5 data.
15    I find the manuscript well written and the structure has a logical flow. I have only minor comments which I hope will help
clarify some of the discussion.

Dear anonymous reviewer #1,
thank you very much for your valuable comments. Please find our point-by-point answers (in blue) to your comments below.
20 Furthermore, Figs. S10 and S11 will be merged and moved from the supplement as new Fig. 9 to the main text. The regression
results therein changed slightly because we now include 50 instead of 30 ensemble members for MPI-ESM-LR (Olonscheck
et al., 2023) and we rearrange the code to be fully compatible with Hall et al. (2021).

Detailed comments:
25

- Abstract: I would suggest expanding the second last paragraph to clarify what is meant. I did not understand this until after reading the manuscript. Then, I would suggest to remove the last sentence of the Abstract, or at least clarify: It is somewhat misleading as no attempt at improving the representation of SPV dynamics is made, and there is not enough discussion of what could be the cause for the observed biases.

To improve clarity of the abstract, we refine the second last paragraph. We also modif the last sentence to be more compatible with our manuscript. The part then will reads: *A good performance in representing the geometric-based diagnostics in rank histograms does not necessarily imply reliability and therefore a good performance in simulating displacements and splits. Assessing the model biases and their representation of SPV dynamics is needed to improve credibility of climate model projections, for example by giving stronger weightings to better performing models.*

- Introduction/Methods: It is argued that one has to use large ensembles to assess wintertime NH variability - which is fine. However, ERA5 is assumed the truth, even though it only represents one ensemble member (the one we call "reality"). I am not suggesting that's a problem, but I would welcome a short discussion on why evaluating large ensembles to one single realisation still makes sense.

We acknowledge your concern regarding the use of ERA5 as the reference dataset, representing a single ensemble member. While ERA5 is widely accepted as a reliable benchmark, we understand your interest in the rationale behind evaluating large ensembles against a single realization.

In response, we emphasize that ERA5 is designated as a state-of-the-art benchmark regarding its constraint by observations and high horizontal and vertical resolution compared to other reanalyses (Hoffmann and Spang, 2022), not an absolute truth. In the future work our results should be supported using multiple reanalysis datasets and analysing different time period to assess stationarity of model performance. However, our study aims to explore the period 1979–2014 covering years when as many observations as possible were assimilated in ERA5 including satellite observations (Hersbach et al., 2020). We recognize the importance of discussing why evaluating large ensembles against a single realization is meaningful, and we will incorporate a concise discussion on this aspect in our revised manuscript.

- It is not clear - neither from Methods nor Code availability - whether the actual code from Seviour et al. (2013) has been used to compute the aspect ratio and centroid lat/lon. Please clarify.

The actual code from Seviour et al. (2013) has been modified accordingly as seen in the updated version of Kuchar and Öhlert (2024). We will mention this explicitly in the revised manuscript.

- Section 3: It would be helpful to provide a sense of how large the biases are. If one is not accustomed to look at rank histograms, it is difficult to judge whether the the shown biases are small or large - that is, whether they are concerning or within acceptable limits. For instance, what does it mean if in Fig. 1c one of the bars peaks around the number 200

but the adjacent bars are below that value? The authors say on line 147 "such strong biases" but I don't know if I should consider them strong for the work I would like to do with those models.

To support the interpretation of rank histograms, we provide bias, spread and the $\chi^2$ statistics, which quantifies how close the rank histogram is to an ideal uniform distribution, for each model and variable above each rank histogram. We will add their description in new Fig. 3 and summarize models' performance in new Tab. S1. Furthermore, we will state in the method section that these statistics should serve in relation to the other models instead of defining any threshold for a good or bad model. Consequently, we go through the manuscript and try to avoid bold statements such as "such strong biases".

– I know it's not the focus of this paper, but some interpretation of results would be great. For instance, models seem to have similar biases at all three levels in centroid latitude, but they have different biases in aspect ratio at different levels. Is there something we can induce or learn from that?

We will provide several discussion instances throughout the manuscript, particularly in Section 5. For instance, l220-225 describe a connection between lower stratospheric winds and SPV split frequency in the CMIP6 models was already noted by Hall et al. (2021) and Wu and Reichler (2020). This could explain larger biases at lower altitudes. However, this explanation is limited to particular models and a more detailed investigation of particular biases is needed. Furthermore, we state that using the rank histograms of aspect ratio at 50 and 100 hPa (see supplementary Figs. S3, S8), the models can be separated into two groups according to their behavior in relation to the results at 10 hPa. One group of models shows larger biases at lower altitudes (CanESM2, INM-CM5-0 and UKESM1-0-LL). In the other model ensembles the bias is weaker at lower altitudes (CanESM5, CESM2, INM-CM5-0, IPSL-CM6A-LR, MIROC6 and MPI-ESMs). We speculate that this is connected with tuning parameters of GW parameterizations as shown in Sigmond et al. (2023).

**3 Ref #2**

GENERAL COMMENTS

The manuscript assess the representation of the geometry of the winter SPV in a few model ensembles, and compares the model ensembles' SSW and SPV frequency to the observed. To persue this analysis, geopotential height data over different stratospheric levels and at daily frequency is taken from the historical simulations of climate models participating in CMIP5 and CMIP6. Based on their results, the authors filter the models that are best at representing the SPV spatial distribution and SSW frequency, and conclude that a good calibration of the SPV shape does not necessarily correspond with an adequate representation of split and displacement SSWs. Outlook on the possible reasons underlying inter-model differences is also provided. Although the topic is of interest for the community and the results might be useful for the institutions developing the different models, I find that the present manuscript is of poor quality. This is because (i) the text is fragmented and difficult to follow, (ii) the language is often imprecise, (iii) the methods are not clearly presented, (iv) the many differences across models

90  and resolutions are difficult to grasp. The whole manuscript needs to be generally and carefully revised before being fit for publication in WCD. In the following, some suggestions that should help improve the afore-mentioned issues are included.

Dear anonymous reviewer #2,

thank you very much for your valuable comments. Please find our point-by-point answers (in blue) to your comments below.

95  Additionally we will go through the entire manuscript to improve presentation quality, make it easier to follow and clarify points that were not adequately described. Furthermore, Figs. S10 and S11 will be merged and moved from the supplement as new Fig. 9. The regression results therein will change slightly because we now include 50 instead of 30 ensemble members for MPI-ESM-LR (Olonscheck et al., 2023) and we rearrange the code to be fully compatible with Hall et al. (2021).

100  SPECIFIC COMMENTS

1. To improve the text please work on fluency and on the precision of the wording. Often adjacent sentences repeat the same concepts, or are fragmented, or badly connected. For example, see some of the points in the TECHNICAL CORRECTIONS.

We will implement the technical corrections and will work on fluency and on the precision of the wording in the whole

105  manuscript.

2. The methods need to be expanded and/or clarified. Specifically, the meaning of the geometric diagnostics can be better explained, the methods behind the rank histogram (RH) and ROC/AUC calculations are not clear.

110  We will provide qualitative descriptions of the moment diagnostics and will add detailed descriptions in the supplement. The methods behind RH and ROC/AUC will be clarified too.

3. It would be useful to compare the list of ERA5 SSW split and displacement events determined with geometric methods (i.e., those identified here) to those identified with more traditional methods (lists of events and their types are available online

115  or in literature). Additionally, please provide the magnitude of the mean SSW frequency for each model, together with the AUC.

Frequency of ERA5 SSW split and displacement events determined with geometric methods is within uncertainty of other methods ($\sim$6.94 events per decade, $\sim$6.66 events per decade including displacement and splits events only, displacement/split ratio equals to 1.4). We provide the list of ERA5 SSW split and displacement events in Tab. R1 below to document this

120  agreement.

We will also show and discuss the mean displacement and split frequency for each model in the y-axes of new Fig. 9 (formerly in the supplement) according to Hall et al. (2021). However, this metric also includes the zonal wind component as other more traditional methods, it is therefore not fully comparable with our AUC plots using only the geopotential height and no

**Table R1.** List of SSWs in ERA5

|    | onset dates | vortex type |
|----|-------------|-------------|
| 0  | 1979-02-22  | split       |
| 1  | 1980-02-29  | displaced   |
| 2  | 1981-03-04  | displaced   |
| 3  | 1981-12-04  | unclassified|
| 4  | 1984-02-23  | displaced   |
| 5  | 1985-01-01  | split       |
| 6  | 1987-01-23  | displaced   |
| 7  | 1987-12-08  | split       |
| 8  | 1988-03-14  | split       |
| 9  | 1989-02-21  | split       |
| 10 | 1998-12-15  | displaced   |
| 11 | 1999-02-25  | split       |
| 12 | 2000-03-20  | displaced   |
| 13 | 2001-02-11  | split       |
| 14 | 2001-12-30  | displaced   |
| 15 | 2002-02-17  | displaced   |
| 16 | 2003-01-18  | split       |
| 17 | 2004-01-05  | displaced   |
| 18 | 2006-01-21  | displaced   |
| 19 | 2007-02-24  | displaced   |
| 20 | 2008-02-22  | displaced   |
| 21 | 2009-01-24  | split       |
| 22 | 2010-02-09  | displaced   |
| 23 | 2010-03-24  | displaced   |
| 24 | 2013-01-06  | split       |

constraints on event separation. Nevertheless, we will include AUC in Tab. R2 as suggested below.

125

4. The Discussion and Outlook section is dispersive, and should be organised more clearly, without mixing your results (discussion and limitations) with the outlooks. Probably, some paragraphs in the Results section would fit better in the Discussion section (e.g., see TECHNICAL CORRECTIONS). Finally, it would be useful to add a table summarising the performances/biases of each model, by geometric diagnosis and by type of SSW event.

130

We will go through the section and will improve its clarity, apply the TECHNICAL CORRECTIONS suggested below and also remove some parts. For several reasons, we refrain from doing a certain suggested restructuring, please see below. We will input the suggested table (see Tab. R2) summarising the models' performance in the supplement.

Table R2: Summary table including metrics: bias, spread and AOC. AOC related to displacement and split events is in the table associated with centroid latitude and aspect ratio, respectively.

| | | 100 hPa | | | 50 hPa | | | 10 hPa | | |
|---|---|---|---|---|---|---|---|---|---|---|
| | | bias | spread | AUC | bias | spread | AUC | bias | spread | AUC |
| CanESM2 | aspect ratio | 590.99 | 213.27 | | 210.74 | 54.06 | | 148.61 | 7.10 | 0.55 |
| | objective area | 343.13 | 122.48 | | 15.73 | 77.34 | | 197.27 | 128.91 | |
| | kurtosis | 11.78 | 226.71 | | 591.70 | 240.18 | | 5525.35 | 3790.66 | |
| | centroid latitude | 418.52 | 79.54 | | 336.71 | 53.66 | | 781.14 | 47.83 | 0.66 |
| | centroid longitude | 118.84 | 66.86 | | 3.13 | 141.43 | | 224.01 | 13.95 | |
| CanESM5 | aspect ratio | 184.78 | 48.57 | | 266.95 | 59.42 | | 772.06 | 140.74 | 0.45 |
| | objective area | 894.65 | 1.51 | | 1520.33 | 23.75 | | 425.06 | 11.87 | |
| | kurtosis | 118.64 | 411.00 | | 16.80 | 353.75 | | 4577.71 | 4368.73 | |
| | centroid latitude | 290.00 | 53.74 | | 276.07 | 65.13 | | 19.30 | 6.20 | 0.59 |
| | centroid longitude | 44.20 | 47.26 | | 1.48 | 15.65 | | 120.33 | 224.55 | |
| CESM2 | aspect ratio | 104.88 | 10.34 | | 271.81 | 32.92 | | 576.90 | 127.07 | 0.59 |
| | objective area | 334.06 | 91.54 | | 476.74 | 166.81 | | 1587.79 | 344.51 | |
| | kurtosis | 6.82 | 217.62 | | 61.09 | 378.66 | | 2239.48 | 2203.39 | |
| | centroid latitude | 252.67 | 88.90 | | 366.22 | 115.06 | | 233.96 | 21.51 | 0.61 |
| | centroid longitude | 26.31 | 50.33 | | 0.01 | 33.15 | | 67.85 | 56.46 | |
| CNRM-CM6-1 | aspect ratio | 6.74 | 0.72 | | 14.67 | 0.03 | | 108.77 | 0.28 | 0.64 |
| | objective area | 3.00 | 5.46 | | 51.57 | 14.76 | | 847.55 | 78.80 | |
| | kurtosis | 227.27 | 10.95 | | 866.92 | 176.20 | | 4018.98 | 2329.24 | |
| | centroid latitude | 0.06 | 2.24 | | 22.69 | 0.35 | | 15.46 | 8.79 | 0.71 |
| | centroid longitude | 0.39 | 4.15 | | 15.80 | 2.72 | | 0.90 | 2.94 | |
| INM-CM5-0 | aspect ratio | 257.00 | 69.24 | | 119.68 | 51.19 | | 0.65 | 1.80 | 0.53 |
| | objective area | 26.65 | 71.31 | | 71.70 | 80.82 | | 165.95 | 436.67 | |
| | kurtosis | 52.26 | 221.07 | | 114.18 | 133.80 | | 3.01 | 25.45 | |
| | centroid latitude | 304.27 | 182.85 | | 391.94 | 148.44 | | 90.28 | 7.54 | 0.67 |

Table R2: Summary table including metrics: bias, spread and AOC. AOC related to displacement and split events is in the table associated with centroid latitude and aspect ratio, respectively.

| | | 100 hPa | | | 50 hPa | | | 10 hPa | | |
|---|---|---|---|---|---|---|---|---|---|---|
| | | bias | spread | AUC | bias | spread | AUC | bias | spread | AUC |
| | centroid longitude | 55.00 | 53.04 | | 0.03 | 75.01 | | 0.52 | 7.36 | |
| IPSL-CM6A-LR | aspect ratio | 41.73 | 4.27 | | 115.30 | 15.54 | | 450.04 | 2.55 | 0.63 |
| | objective area | 507.80 | 75.51 | | 137.37 | 71.86 | | 570.49 | 730.88 | |
| | kurtosis | 1.26 | 94.65 | | 10.65 | 132.02 | | 45.86 | 69.42 | |
| | centroid latitude | 44.41 | 18.49 | | 64.22 | 46.14 | | 156.46 | 7.66 | 0.63 |
| | centroid longitude | 77.08 | 41.45 | | 3.29 | 403.83 | | 289.33 | 6.17 | |
| MIROC6 | aspect ratio | 8.38 | 7.86 | | 6.06 | 13.84 | | 19.07 | 6.55 | 0.59 |
| | objective area | 841.65 | 420.28 | | 900.00 | 790.83 | | 624.89 | 1015.18 | |
| | kurtosis | 62.65 | 0.38 | | 19.48 | 15.54 | | 40.58 | 19.20 | |
| | centroid latitude | 0.16 | 19.33 | | 97.24 | 43.67 | | 332.99 | 19.25 | 0.62 |
| | centroid longitude | 101.61 | 9.90 | | 26.33 | 199.91 | | 209.12 | 4.14 | |
| UKESM1-0-LL | aspect ratio | 99.95 | 19.77 | | 91.16 | 17.09 | | 0.33 | 0.27 | 0.58 |
| | objective area | 0.15 | 127.42 | | 161.29 | 91.84 | | 67.10 | 14.83 | |
| | kurtosis | 54.18 | 107.00 | | 17.97 | 51.80 | | 421.63 | 99.53 | |
| | centroid latitude | 1.94 | 0.19 | | 18.81 | 1.03 | | 16.88 | 27.50 | 0.66 |
| | centroid longitude | 0.33 | 5.42 | | 1.19 | 3.25 | | 156.71 | 29.47 | |
| MPI-ESM1-2-HR | aspect ratio | 1.81 | 2.84 | | 11.33 | 1.06 | | 116.15 | 0.01 | 0.56 |
| | objective area | 535.27 | 72.35 | | 484.98 | 196.42 | | 1046.07 | 824.85 | |
| | kurtosis | 3.57 | 8.82 | | 27.34 | 78.91 | | 453.43 | 40.79 | |
| | centroid latitude | 151.91 | 10.35 | | 89.40 | 7.42 | | 123.33 | 0.91 | 0.59 |
| | centroid longitude | 16.50 | 9.09 | | 1.26 | 13.32 | | 4.53 | 80.94 | |
| MPI-ESM1-2-LR | aspect ratio | 3.77 | 0.56 | | 11.93 | 0.55 | | 242.95 | 7.78 | 0.65 |
| | objective area | 1030.16 | 372.84 | | 776.32 | 568.10 | | 1118.40 | 1266.02 | |
| | kurtosis | 1.91 | 13.73 | | 0.04 | 71.31 | | 324.23 | 102.14 | |
| | centroid latitude | 298.72 | 32.05 | | 154.32 | 12.51 | | 199.48 | 21.33 | 0.66 |
| | centroid longitude | 0.65 | 57.89 | | 26.56 | 2.04 | | 39.20 | 214.11 | |
| GFDL-CM3 | aspect ratio | 271.25 | 116.71 | | | | | | | |

Table R2: Summary table including metrics: bias, spread and AOC. AOC related to displacement and split events is in the table associated with centroid latitude and aspect ratio, respectively.

| | 100 hPa | | | 50 hPa | | | 10 hPa | | |
| --- | --- | --- | --- | --- | --- | --- | --- | --- | --- |
| | bias | spread | AUC | bias | spread | AUC | bias | spread | AUC |
| objective area | 6615.54 | 2364.37 | | | | | | | |
| kurtosis | 42.21 | 423.25 | | | | | | | |
| centroid latitude | 471.85 | 282.04 | | | | | | | |
| centroid longitude | 69.90 | 52.72 | | | | | | | |

5. How does the map of the mean gh10 climatological bias look like for each model? Is it relatable to the more specific rank histogram results? If so, such a figure could be included at the beginning of the Results section.

We show the geopotential height climatology at 10 hPa (gh10) of all analyzed model ensembles and ERA5 for the period 1979–2014 in Fig. R1. It shows that there are models which apparently struggle in simulating gh10 comparable to ERA5 (CanESMs, CESM2). On the other hand, visual resemblance between ERA5 and UKESM1-0-LL corresponds to our indications as best-performing model. However, details about reliability of recent large-ensemble model simulations in representing the SPV and its spatial variability cannot be fully decomposed based on this figure. Thank you for the comment, we will include the figure to the manuscript in the Method section and use it to motivate the moment diagnostics for analysis of spatial variability.

6. It would be useful to show an example of a ROC curve for one model, displaying both the split and displacement curve. This could go in the Methods or in the Results, and would be useful for explaining how the ROC curve is constructed.

Since we display all ROC curves in the supplement (see Figs. S1 and S2), we also explain the ROC curves in the supplement. As an example we include Fig. R2 to show ROC for displacements and splits. Fig. R2**B** demonstrates that CanESM5 cannot discriminate split events better than random guessing. We mention it in the Methods and in the Results sections.

7. Since you mention the diagnostics of the vertical structure of the SPV in the abstract, it is advisable to show a couple of meaningful plots (or statistics values) for the other stratospheric levels in the main manuscript.

Thank you for the advise. We will include two new figures (R1, R2) and merge and move Figs. S10 and S11 from the supplement as new Fig. 9. Thus, we decided though to keep figures for 50 and 100 hPa in the supplement, since their discussion is overall limited, to keep the manuscript concise.

8. You should mention any previous use and results of geometric diagnostics in your Introduction section. Or did I miss this? On the other hand, SPV projections are not your main focus, so you should mention them briefly to motivate your work or more extensively only if directly related.

[Figure]

**Figure R1.** Geopotential height climatology at 10 hPa (gh10) of all analyzed model ensembles: CanESM2 (**A**), CanESM5 (**B**), CESM2 (**C**), CNRM-CM6-1 (**D**), INM-CM5-0 (**E**), IPSL-CM6A-LR (**F**), MIROC6 (**G**), UKESM1-0-LL (**H**), MPI-ESM1-2-HR (**I**) and MPI-ESM1-2-LR (**J**), and ERA5 for the period 1979–2014. The black and purple line represents contour of 30000 m in ERA5 and in a particular model, respectively.

[Figure]

**Figure R2.** Receiver operating characteristics (ROC) curves for displacements (**A**) and splits (**B**) in CanESM5. Bin values indicated along ROC are probabilities whether model simulates displacements and splits across its ensemble members, respectively. Area under the ROC curve is visualized and also specified in the figure. The dashed line represents random disrcimination skill, i.e. AUC=0.5.

There are several references (Charlton and Polvani, 2007; Mitchell et al., 2013; Seviour et al., 2013, 2016; Hall et al., 2021) in the Introduction section and in Section 2.2 (Matthewman et al., 2009; Seviour et al., 2013). We also list a recent review on SSWs (Baldwin et al., 2021) where additional references can be found. In our opinion, the list is quite extensive. Furthermore, the discussion and abstract related to SPV projections will be adjusted to the following: *Assessing the model biases and their representation of SPV dynamics is needed to improve credibility of climate model projections, for example by giving stronger weightings to better performing models.*

TECHNICAL CORRECTIONS

General Please check the usage of the adjectives 'low'/'high' (e.g. related to bias), and replace with 'weak'/'strong' or 'negative'/'positive' when ambiguous.

We will go through the manuscript and make appropriate changes.

Abstract Line 1 : « each hemisphere ».

Corrected.

Line 1 : The word « phenomenon » is not adequate for describing the SPV. Please change.

Modified accordingly.

Lines 5-13 : Abstract would gain in clarity if the description was separated into geometric diagnosis and split-displacement diagnosis (e.g. line 6 shifted after the geometric diagnosis). The fluency of the text would improve.

180    Modified accordingly.

Introduction Line 25 : « relates ».
*relates* replaced with *relate*.

185    Lines 25-26 : The term « windiness » is ambiguous. I would suggest to replace the first two elements in the list with « the position of the jet and the precipitation patterns over Europe and the Mediterranean region ».
Modified accordingly.

Line 27 : « where the SPV is highly variable ».
190    We added *where*.

Line 31 : « from a final warming».
We skip the sentence since it is irrelevant.

195    Lines 32-33 : The sentence is difficult to read, could you please reformulate ?
We skip the sentence since it is irrelevant.

Lines 37-39 : The sentence is out of place in the present position. Please fit adequately in the discussion of literature.
We skip the sentence since it is discussed before.
200

Lines 57-58 : Do you mean that a limitation of the single-member analysis is that one can't distinguish between natural and inter-model variability ? This is not the meaning emerging from the sentence, please correct or clarify.
Modified accordingly.

205    Line 63 : « will be is », for consistency of the verb tense.
Corrected.

Lines 63-63 : The following sentence/content should be shifted in an adequate position, before the outline. « Due to the large SPV variability and the high SSW frequency in the NH, we limit our analysis to the NH SPV. »
210    We move the sentence accordingly.

Methods Lines 70-72 : You can remove the reasoning here, as it is already discussed in the Introduction.
Done.

215 Line 76 : Please change to « climate models used in our analysis are described in table 1 ».

Modified accordingly.

Line 78 : Inconsistency between the two sentences. Please modify.

Modified accordingly.

220

Line 80 : The first part of the line is accessory, please remove or include in compact form as part of the previous sentence.

Done.

Line 91-92 : The last sentence gives a qualitative description of the different moment diagnostics, but the link with the
225 diagnostics (two sentences back) is not obvious – I suggest giving this information just following the list of diagnostics.
Moreover, an extension of their description, here or at the beginning of each section in the results, would be welcome for any
non-expert reader.

We will provide qualitative descriptions of the moment diagnostics in the text and more detailed descriptions including their
derivations in the supplement.

230

Lines 95-99 : The description of rank histograms (RH) is confusing. Please describe precisely and clearly how the counts
within each histogram are determined, and why the distribution should be uniform in a well-calibrated forecast. Is the sentence
« The histogram counts of all bins greater or equal k are then increased by one » correct? From what I read from Sect. 2 in
Hamill et al. 2001 (your ref.), only one of the bins is updated at each timestep, corresponding to the forecast ensemble bin
235 where the observation falls.

Thanks for your notice. We improved the description of RH in this parahraph accordingly.

Line 119 : Mention where and how the perfect model range method is used within the present work.

We will mark the description in the Section 2.4. The details of calculation can be found at our Github repository (Kuchar
240 and Öhlert, 2024).

Line 122-24 : It is not clear to me if your definition of displacements and splits corresponds to the description you give in
these lines. Could you make it more explicit?

Done.

245

Line 126 : Insert ref. for ROC and AUC.

See former l125-126.

Line 127 : Can you describe in this paragraph how the thresholds for drawing the ROC curve are computed ? And what you mean by true and false positive rates in a historical simulation (rather than in a forecast) ? An example of a ROC curve would also be welcome, including descriptive labels for x and y axes.

We described ROC in details in this paragraph and we also included Fig. R2 as an example.

Results Line 139 : It would be useful to specify again, within this sentence, that high values of the aspect ratio correspond to split events.

We will revise the introduction to the section to provide easier access to the topic for the reader. The part then reads: *The aspect ratio of the SPV is determined by the ratio of the major to the minor ellipse axis. Thus, it measures how stretched the SPV is providing high values for stretched and low values for more circular SPVs. Exceptionally high aspect ratio values are of particular interest for SPV dynamics as they are often associated with SPV disturbances, in particular indicating splitting events (Seviour et al., 2013).*

Line 143 : No need to repeat the aside « which is considered as ground truth for the analysis ».
Will be deleted as redundant.

Line 144 : « are biased simulating simulate ».
Modified to *are biased and simulate*.

Line 145 : « high positive ».
*positive* added.

Line 146 : How is the significance of the bias computed ?
We will omit the word *significant* and use *considerable* instead.

Lines 148-49 : You could merge the last two sentences of the paragraph, there is a partial overlap.
Done.

Line 152 : Does « low » mean 'weak' or 'negative' ?
*low* replaced by *weak*.

Line 153 : « model ensembles ».
Added.

Line 165 : Please specify pressure levels each time.

Specified.

285

Line 192 : « high » means positive ?

*high* replaced with *strong*.

Lines 200-01 : I don't understand the sentence starting by « However.. ». Please clarify or remove.

290    We delete this sentenced as discussed in Section 5.

Lines 212-14 : Please specify large objective area or strong SPV, or describe what large SPV means. Check the same in the rest of the subsection.

We delete *small SPV* from l211. We also remove the following sentence: *This combination results in a strong underestima-*

295    *tion of a large SPV occurence.*

Lines 221-228 : These lines are better fitted for the Discussion section.

In Section 5 we discuss this aspect more generally in relation to other model parameters such as model resolution and GW parameterizations and their tuning parameters.

300

Line 240 : Specify that the assertion is valid for displacements. Or else present only the displacement results in the first introductory sentence.

Will be done by including For displacements,...' at the beginning of the sentence and additionally we add clarification of what can be seen in the 2 panels A and B (displacements and splits).

305

Lines 242-44 : I find the discussion quite vague. Could you clarify?

We will improve clarity and rearrange the whole section 4 according to new Fig. 9.

Sections 4.1 and 4.2 : The second part of each subsection is better fitted for the Discussion section, specially when the

310    manuscript's results are compared with previous literature.

We refrain from moving these parts to the discussion section for several reasons. Firstly, we feel that a brief placement of the results (with literature) is fine for the results section, also to leave room in the discussion section for the more detailed discussions, and to not make the discussion section too lengthy. Secondly, we will include the scatter plots to the main text, to improve the results section, and that is connected with the mentioned parts, and would therefore lead to the discussion section

315    having new results, which is not very pretty usually.

Lines 314-17 : These lines are rather vague and unrelated from the results and conclusions of the manuscript. Please remove if not otherwise justified.

We rearrange the paragraph to improve clarity and make the sentences relevant to previous statements.

320

Line 323 : It would be good to acknowledge that you are not analysing or describing GW parametrisation in this work.

We acknowledge that we do not demonstrate how the GW drag differences project to the intermodel differences in the SPV dynamics. For further discussion we refer to Hájková and Šácha (2023) and Sigmond et al. (2023). The part will read:

*Recently, Hájková and Šácha (2023) showed that the SPV climatologies in CMIP6 models are largely insensitive to high lati-*

325 *tude wave drag, but also state that SSW simulation can be sensitive to small nuances in model dynamics. Dedicated analyses are needed to fully assess the impact of various wave drag mechanisms on SPV geometry and SSWs, including consideration of the non-linear feedbacks between wave drag and mean flow. For example, Sigmond et al. (2023) have attributed the difference in simulated SSW frequency between CanESM2 and CanESM5 to changes in settings of gravity wave tuning.*

330 Line 328 : Is this phrased correctly ? If so, it is difficult to understand what you are concluding.

The sentence will be rearranged to the following: *Therefore, these uncertainties in the models may be attributed to different gravity wave parameterizations (Sigmond and Shepherd, 2014).*

Lines 348-352 : This discussion is too general – it should connect more with the results of your work if you want to include

335 it.

The discussion will be removed.

Conclusions Line 360 : What do you mean by « stability »?

We omit *stability* in the revised manuscript.

340

Line 370-71 : Can you tell why your results are different from previous literature ? Is it a different metrics?

Hall et al. (2021) use the preindustrial control simulation of each available model from $CMIP^6, which has a prescribed or calculated CO_2$ concentration, designed for the evaluation of unforced variability. They also use slightly diferent set of models. We rearrange our code for detections of SSWs to be fully compatible with Hall et al. (2021). Therfore, our results for displacement SSWs in

345 new Fig. 9**A** are in agreement with Fig. 3 in Hall et al. (2021). In case of split SSWs, our results are more dispersed compared to Hall et al. (2021). Unlike their results, the reanalysis values lie within the 95% confidence interval of the OLS fit. As we can rule out that the size of ensemble members might not be sufficiently large in our study, we argue that the fit is not so robust for split SSWs because stretching tendency of polar vortex is accompanied with the centroid-latitude tendency to equatorward values (Mitchell et al., 2011).

350

Line 375 : At some point you had some contradictory results regarding resolution and SPV representation (see e.g. Line 199). Is it worth taking this into account in the present assertion?

We conclude that models with a finer vertical resolution generally simulate the SPV and SSWs better with reference to ERA5. However, the sentence regarding the MPI-ESM1-2 simulations will be modified in the following: *Both MPI-ESM1-2 simulations contains bias but dome-shaped RH in MPI-ESM1-LR indicates larger ensemble spread.*

Line 380 : « variability » instead of « behaviour ».
Thanks, will be done.

Line 380 : Replace « set the basis » with « can constitute/be a reference ».
Thanks, will be done.

Line 380 : Rephrase « with individual models changing individual model components ».
Will be rephrased to: ...our findings can be a reference for studies testing the sensitivity of individual models to particular model components or settings therein, such as...

Line 382 : « the SPV spatial variability ».
Thanks, will be done.

Line 383 : Please rephrase « for finding the adjustment screws to improve their performance ».
Will be rephrased to: "tuning and calibration to improve their performance"

Figures Figure 1-5 : Increase font size of model name and statistics above each plot. Mention the different models and the respective letter labels in the caption.
Thanks for your suggestion, will be done.

**References**

[revised manuscript text omitted]

---

## Author Response (AR1)

Dear all,

we enclosed our final replies to both referees in the open discussion as AC3 on January 26, 2024. You can find our point-by-point response to the reviews including a list of all relevant changes made in the manuscript here.

With regards on behalf of all authors

Ales Kuchar

---

## Referee Report (RR1)

The manuscript is substantially improved in terms of text, language and clarity. However, my recommendation for publication is conditional on overcoming my first major comment. A list of major and specific comments, referring to the line numbering in the marked-up version of the manuscript, follows.

MAJOR COMMENTS
1. My greatest concern regards the ROC curve analysis, since, from the explanations in the Methods and Supplementary Material, I still fail to understand how you build and interpret it. In the Supplementary you say « the false positive rate indicates the probability that the ensemble is simulating an event, even though there was no event observed in the reanalysis ». Do you check this day by day, or year by year, or using some other selection criterion ?
I have in mind contingency tables for ensemble forecasts, where false positive rate corresponds to the ratio of forecasted positives in a period of time when a negative realisation was observed, the true positive rate is the ratio of forecasted positives when a positive realisation was observed, but cannot imagine what this represents for a climate simulation.
In fact, in ensemble climate simulations, differently from initialised forecasts, you do not expect an exact temporal correspondence between simulated and reanalysis events (splits and displacements) because of internal climate and weather variability.
So, what do you mean by true and false positive rates in this case ? How do you obtain the ROC curve, in the detail of the computation ? What does it mean and how do you interpret it ?
And based on your answers to these questions, how are you then able to discuss over and under-representation of SPV displacements and splits, and the model performance in distinguishing between SPV displacements and non-displacements ?
Because of my not understanding the above, all of Section 4 is still unclear to me.

If I may add a suggestion, a much less convoluted way of understanding the model fidelity in the frequency of split and displacement SSWs would be to compute the monthly frequency of the events across models and compare this with the reanalysis frequency. Hopefully, relations with the SPV diagnostics might become more apparent.

2. Section 3 would benefit from a filtering of the most important information. The many details are difficult to follow.

3. In the Abstract and in the Conclusions I suggest to remove the technical terms of the SPV diagnostics and to give an intuitive interpretation of the result. A reader should understand the main conclusions without having to browse the details of the Methods section.

4. In Table S1 you could evidence, for each diagnostic, the model/s with the weakest best performance, and refer to the table in the Conclusions Section.

TECHNICAL CORRECTIONS

Line 9 : You should explain the results without mentioning the technical names of the different diagnostics, since these haven't been defined yet. (See Maj. Comm. 3)

Line 34 : « mid-latitude zonal mean zonal wind reverses... ».

Line 38 : « into two separate vortices ».

Line 74 : Shift and merge sentence with that in line 67, preceeding the outlook paragraph.

Line 92 : Remove « , or in other words, ERA5 data serves as ground truth in our analysis. ». The sentence is not useful and out of place.

Line 94 : As in previous comment, remove « however, clearly not as an absolute truth ».

Line 96 : replace with « when satellite observations were assimilated in ERA5. ».

Lines 104-112 : organise in a list of bullet points.

Paragraph 117 : Shift this at the beginning of the Results' section. It is out of place in the Methods.

Line 128 : I repeat a comment from my previous review. Is the sentence « The histogram counts of all bins greater or equal k are then increased by one » correct? From what I read from Sect. 2 in Hamill et al. 2001 (your ref.), only one of the bins is updated at each timestep, corresponding to the forecast ensemble bin where the observation falls.

Line 133 : Not clear. Please be more precise in the wording.

Line 140 : How is the U-shape (spread) computed ?
The use of the word « spread » is confusing in this context, as it reminds of ensemble spread, which is not the case here. You could use « spread bias » or « dispersion bias » instead, here and in the rest of the manuscript. You could also mention that 0 values in the statistics indicate perfect model dispersion (if I understood correctly !).

Line 172 : What is the methodology from Hall et al. (2021) ?? Please specify.

Line 176 and table S2: Please specify in the text and caption if this is the SSW list relative to your definitions, or if it's taken from literature.
The interesting point would be to show two lists of dates and types, yours and that used in literature, and discuss the differences.

Line 184-187 : Make shorter – you should summarise in a few words the Methods' description at the moment.

Line 251-255 : condence UKMO model results, and say in clearer language.

Line 264-66 : I am not convinced by this sentence. Remove if not adequately motivated.

Line 274 : replace « too » with « anomalously ».

Lines 276-277 : replace « small » with « weak » and « large » with « strong ».

Lines 385-401 : The discussion is quite dispersive, because of alternating from literature to your own results multiple times. I would suggest to separate the two clearly within the paragraph.

Lines 402-413 : Very long discussion on a topic that is outside your scope. You should filter and compress this paragraph.

---

## Author Response (AR2)

**Replies to reviews for:**
**Large ensemble assessment of the Arctic stratospheric polar vortex**

WCD, Ales Kuchar et al.

**1 Reply to reviews**

Dear Daniela Domeisen,

thank you very much for helping in the publishing process of this paper. We appreciate the helpful comments of the reviewer
**1. We revised the manuscript accordingly, such that we hope it can then be published in WCD. Please find our point-by-point**
answers to the comments below. Furthermore, we changed the title according to your own suggestion.

Best wishes

Ales, Roland, Maurice and Christoph

**2 Ref #1**

Dear anonymous reviewer #1,
thank you very much for your valuable comments. Please find our point-by-point answers (in blue) to your comments below.

**2.1 MAJOR COMMENTS**

1. My greatest concern regards the ROC curve analysis, since, from the explanations in the Methods and Supplementary
Material, I still fail to understand how you build and interpret it. In the Supplementary you say «the false positive rate indicates
the probability that the ensemble is simulating an event, even though there was no event observed in the reanalysis». Do
you check this day by day, or year by year, or using some other selection criterion? I have in mind contingency tables for
ensemble forecasts, where false positive rate corresponds to the ratio of forecasted positives in a period of time when a negative
realisation was observed, the true positive rate is the ratio of forecasted positives when a positive realisation was observed,
but cannot imagine what this represents for a climate simulation. In fact, in ensemble climate simulations, differently from
initialised forecasts, you do not expect an exact temporal correspondence between simulated and reanalysis events (splits and
displacements) because of internal climate and weather variability. So, what do you mean by true and false positive rates in

25 this case? How do you obtain the ROC curve, in the detail of the computation? What does it mean and how do you interpret it? And based on your answers to these questions, how are you then able to discuss over and under representation of SPV displacements and splits, and the model performance in distinguishing between SPV displacements and non-displacements? Because of my not understanding the above, all of Section 4 is still unclear to me.

If I may add a suggestion, a much less convoluted way of understanding the model fidelity in the frequency of split and
30 displacement SSWs would be to compute the monthly frequency of the events across models and compare this with the reanalysis frequency. Hopefully, relations with the SPV diagnostics might become more apparent.

We check the forecast and observation probability of crossing split or displacement thresholds for daily data of the months from November through March in the NH. The detailed procedure how Fig. 2 and therefore true positive and false positive rates using a set of increasing probability bins from the upper right corner to the lower left corner are calculated can be found
35 in the notebook ROC_values.ipynb (Kuchar and Öhlert, 2024). As shown therein the dichotomous contingency are calculated along the probabilities of whether the model simulates displacements or splits across its ensemble members. Therefore, there is no assumption about the temporal correspondence of these events but the probability distributions between ERA5 and model ensembles are assessed. To avoid that for readers these points are unclear, we added this sentence to the paragraph dedicated to the ROC curve in the manuscript.

40 As explained in the manuscript, we interpret higher AUC (arc of the corresponding ROC curve more closely to the upper left corner) as a model ensemble discriminates split or displacement SSW events better than a model ensemble with lower AUC (arc of the corresponding ROC curve more closely to the diagonal line). For further details about the application and interpretation of the ROC diagram we refer to Wilks (2011) or Jolliffe and Stephenson (2012). We added this sentence to the paragraph dedicated to the ROC curve in the supplement.

45 For example, SPV displacements are less frequent in models that simulate a bias towards higher values of centroid latitude. However, the ROC diagram is insensitive to certain types of biases (Kharin and Zwiers, 2003), as a biased model may still have good statistical resolution (e.g. CESM2). The ROC can still be considered as a potential skill (usefulness) when the model is correctly calibrated (Wilks, 2011). This fact can explain the differences between CanESM2 and CanESM5. While CanESM2 distinguishes between displacement events and stable SPV conditions better than CanESM5 according to the AUC, CanESM5
50 performs much better in the rank histogram analysis for the centroid latitude at 10 hPa.

Thanks for your suggestion to compute the monthly frequency of the events across models and compare this with the frequency of reanalysis. Nevertheless, we assess the model fidelity in the frequency of split and displacement SSWs in Fig. 9. The related discussion can be found in Section 4. Based on this, we concluded that the geometric SPV biases indicates also biases in split and displacement frequency as in Seviour et al. (2016) and Hall et al. (2021). However, this does not necessarily imply
55 that bias-corrected models simulate split and displacement frequencies according to the reanalyses (Wu and Reichler, 2020).

2. Section 3 would benefit from a filtering of the most important information. The many details are difficult to follow.

Sect. 3 mostly is a mere description of the analysis results. Obviously not all points mentioned here are discussed later in the paper, but we do aim at a certain degree of completeness in this results description. As we already performed a fair bit of

condensing of this section in the process of the paper writing and the reviewers statement is very unspecific, we do not see the need for action here. Moreover, the reviewer was asking for additional descriptions and analyses for this section in the first round of reviews, which makes this statement appear somewhat ambiguous.

3. In the Abstract and in the Conclusions I suggest to remove the technical terms of the SPV diagnostics and to give an intuitive interpretation of the result. A reader should understand the main conclusions without having to browse the details of the Methods section.

We enriched the technical terms in the abstract and in the conclusions section with intuitive descriptions, or removing bits, or extending with the intuitive explanations.

4. In Table S1 you could evidence, for each diagnostic, the model/s with the weakest best performance, and refer to the table in the Conclusions Section.

Done.

**2.2 TECHNICAL CORRECTIONS**

Line 9:You should explain the results without mentioning the technical names of the different diagnostics, since these haven't been defined yet. (See Maj. Comm. 3)

Done, see answer to Maj. Comm. 3

Line 34: «mid-latitude zonal mean zonal wind reverses...».

Corrected accordingly.

Line 38: «into two separate vortices».

Corrected accordingly.

Line 74: Shift and merge sentence with that in line 67, preceeding the outlook paragraph.

Modified accordingly.

Line 92: Remove «, or in other words, ERA5 data serves as ground truth in our analysis.». The sentence is not useful and out of place.

Removed accordingly.

Line 94:As in previous comment, remove «however, clearly not as an absolute truth».

Removed accordingly.

95     Line 96: replace with «when satellite observations were assimilated in ERA5.». Lines 104-112: organise in a list of bullet points.

Replaced with:

'...when, among others, satellite observations were assimilated to ERA5...'

100     Paragraph 117: Shift this at the beginning of the Results' section. It is out of place in the Methods.

Corrected accordingly.

Line 128: I repeat a comment from my previous review. Is the sentence « The histogram counts of all bins greater or equal k are then increased by one » correct? From what I read from Sect. 2 in Hamill et al. 2001 (your ref.), only one of the bins is 105 updated at each timestep, corresponding to the forecast ensemble bin where the observation falls.

Thanks for your comment. We modified the sentence accordingly.

Line 133: Not clear. Please be more precise in the wording.

We modified the sentence to increase clarity.

110

Line 140: How is the U-shape (spread) computed? The use of the word «spread» is confusing in this context, as it reminds of ensemble spread, which is not the case here.You could use «spread bias» or «dispersion bias» instead, here and in the rest of the manuscript. You could also mention that 0 values in the statistics indicate perfect model dispersion (if I understood correctly!).

The spread is computed according to the methodology by Jolliffe and Primo (2008). The code that was used to produce all 115 plots in this study is available via Zenodo (Kuchar and Öhlert, 2024). We modified the sentence so that the context is clear.

Line 172: What is the methodology from Hall et al. (2021)?? Please specify.

We modified "the methodology from Hall et al. (2021)" to "the methodology to detect displacement and split SSWs from Hall et al. (2021)".

120

Line 176 and table S2: Please specify in the text and caption if this is the SSW list relative to your definitions, or if it's taken from literature. The interesting point would be to show two lists of dates and types, yours and that used in literature, and discuss the differences.

As mentioned in the previous sentence, it is determined with the method of Hall et al. (2021). We add "Using this method-125 ology," in the beginning of l176. As we state in the manuscript, this method are within the uncertainty of other methods.

Line 184-187: Make shorter – you should summarise in a few words the Methods' description at the moment.

We removed the sentence:

"Exceptionally high aspect ratio values are of particular interest for SPV dynamics as they are often associated with SPV dis-turbances, in particular indicating splitting events (Seviour et al., 2013)."

Line 251-255:condence UKMO model results, and say in clearer language.

Condensed and made clearer.

Line 264-66: I am not convinced by this sentence. Remove if not adequately motivated.

We modified the sentence providing a reasoning.

Line 274: replace «too» with «anomalously».

Replaced accordingly.

Lines 276-277: replace «small» with «weak» and «large» with «strong».

Replaced accordingly.

Lines 385-401: The discussion is quite dispersive, because of alternating from literature to your own results multiple times. I would suggest to separate the two clearly within the paragraph.

Through minor restructuring we made the paragraph clear and non-dispersive.

Lines 402-413:Very long discussion on a topic that is outside your scope.You should filter and compress this paragraph.

We have compressed this part so that it meets the reviewers taste for such a discussion.

**References**

Hall, R. J., Mitchell, D. M., Seviour, W. J., and Wright, C. J.: Persistent model biases in the CMIP6 representation of stratospheric polar vortex variability, Journal of Geophysical Research: Atmospheres, 126, e2021JD034 759, https://doi.org/10.1029/2021JD034759, 2021.

Jolliffe, I. T. and Primo, C.: Evaluating rank histograms using decompositions of the chi-square test statistic, Monthly Weather Review, 136, 2133–2139, https://doi.org/10.1175/2007MWR2219.1, 2008.

Jolliffe, I. T. and Stephenson, D. B.: Forecast verification: a practitioner's guide in atmospheric science, John Wiley & Sons, 2012.

Kharin, V. V. and Zwiers, F. W.: On the ROC Score of Probability Forecasts, Journal of Climate, 16, 4145 – 4150, https://doi.org/https://doi.org/10.1175/1520-0442(2003)016<4145:OTRSOP>2.0.CO;2, 2003.

Kuchar, A. and Öhlert, M.: VACILT/reliability_LE: Fourth release of our code repository related to reliability of large ensembles, https://doi.org/10.5281/zenodo.10620326, 2024.

Seviour, W. J., Mitchell, D. M., and Gray, L. J.: A practical method to identify displaced and split stratospheric polar vortex events, Geophysical Research Letters, 40, 5268–5273, 2013.

Seviour, W. J., Gray, L. J., and Mitchell, D. M.: Stratospheric polar vortex splits and displacements in the high-top CMIP5 climate models, Journal of Geophysical Research: Atmospheres, 121, 1400–1413, 2016.

Wilks, D. S.: Statistical methods in the atmospheric sciences, vol. 100, Academic press, 2011.

Wu, Z. and Reichler, T.: Variations in the frequency of stratospheric sudden warmings in CMIP5 and CMIP6 and possible causes, Journal of Climate, 33, 10 305–10 320, https://doi.org/10.1175/JCLI-D-20-0104.1, 2020.

---

## Author Response (AR3)

Dear Dr. Domeisen,

thank you very much once again for helping in the publishing process of this paper. We appreciate the list of technical corrections, which we have implemented accordingly. Furthermore, we managed to do proofreading on our own to avoid major grammar mistakes and typos. In our opinion, the manuscript should be understandable and reads well.

Best wishes

Ales, Roland, Maurice and Christoph